# Robust Multi-Agent Reinforcement Learning with Stochastic Adversary

**Ziyuan Zhou**[1]  **Guanjun Liu**[1]  **MengChu Zhou**[2][3]  **Weiran Guo**[1]

## Abstract

The performance of models trained by Multi-Agent Reinforcement Learning (MARL) is sensitive to perturbations in observations, lowering their trustworthiness in complex environments. Adversarial training is a valuable approach to enhance their performance robustness. However, existing methods often overfit to adversarial perturbations of observations and fail to incorporate prior information about the policy adopted by their protagonist agent, i.e., the primary one being trained. To address this important issue, this paper introduces Adversarial Training with Stochastic Adversary (ATSA), where the proposed adversary is trained online alongside the protagonist agent. The former consists of Stochastic Director (SDor) and SDor-guided generaTor (STor). SDor performs policy perturbations by minimizing the expected team reward of protagonists and maximizing the entropy of its policy, while STor generates adversarial perturbations of observations by following SDor's guidance. We prove that SDor's soft policy converges to a global optimum according to factorized maximum-entropy MARL and leads to the optimal adversary. This paper also introduces an SDor-STor loss function to quantify the difference between a) perturbations in the agent's policy and b) those advised by SDor. We evaluate our ATSA on StarCraft II tasks and autonomous driving scenarios, demonstrating that a) it is robust against diverse perturbations of observations while maintaining outstanding performance in perturbation-free environments, and b) it outperforms the state-of-the-art methods.

## 1. Introduction

Deep reinforcement learning leverages the representational power of deep learning alongside the decision-making capabilities of reinforcement learning, allowing agents to learn and adapt through interactions with dynamic environments (Sutton & Barto, 2018). In Multi-Agent Reinforcement Learning (MARL), agents are not isolated because they actively interact with and adapt to environments in both cooperative and/or competitive frameworks. This interaction enables them to develop their sophisticated strategies and allows them to optimize their behaviors not only on an individual level but also on the system level, thereby enhancing their ability to handle complex tasks. MARL proves particularly valuable for data-driven applications (Cai et al., 2023; Yuan et al., 2023a) like autonomous driving (Wang et al., 2024a; Zhang et al., 2023) and recommendation systems (Deng et al., 2023; Gui et al., 2019), where adaptive and decentralized decision-making is crucial. However, models trained by deep learning often lack robustness to dynamic and noisy environments because they are typically trained under the assumption of independent and identically distributed training data (Goodfellow et al., 2014). This assumption is rarely met in real-world scenarios, leading to performance degradation when models encounter a data distribution that differs from the training set's (Wang et al., 2024b; Zhou et al., 2024a).

In MARL, a deep neural network module exacerbates this robustness issue, causing them to make incorrect or suboptimal decisions in critical situations. It is essential to enhance the robustness of MARL-trained models. Much progress has been made on doing so in deep learning for classification tasks such as adversarial training (Goodfellow et al., 2014; Madry et al., 2018; Wong et al., 2020) where the goal of the adversary is to identify and use the strongest adversary. Yet directly applying these approaches to Single-Agent Reinforcement Learning (SARL) scenarios introduces some great challenges: 1) **Mismatch between short-term and long-term goals**. Aversarial training like the Fast Gradient Sign Method (FGSM) (Goodfellow et al., 2014) and projected gradient descent (Madry et al., 2018), borrowed from classification tasks, perturb the agent's observation at each time step (Huang et al., 2017). However, one-step misclassification does not equate to minimized long-term rewards. 2) **Difficulty in high-**

---

[1]The School of Computer Science and Technology, Tongji University [2]The School of Information and Electronic Engineering, Zhejiang Gongshang University [3]The Helen and John C. Hartmann Department of Electrical and Computer Engineering, New Jersey Institute of Technology. Correspondence to: Guanjun Liu <liuguanjun@tongji.edu.cn>.

*Proceedings of the 42nd International Conference on Machine Learning*, Vancouver, Canada. PMLR 267, 2025. Copyright 2025 by the author(s).

**dimensional environments**. Existing studies (Zhang et al., 2020; 2021a) prove that training an adversary to generate an adversarial perturbation of observation can be modeled as a Markov decision process, in which the reward received by the adversary is the negative counterpart of the reward received by the protagonist agent, and its action space is the observation space of the protagonist agent. Then they use reinforcement learning to train an adversary. However, high-dimensional action space in reinforcement learning presents great implementation difficulties. The work in (Sun et al., 2022b) attempts to address these challenges by training adversaries by using policy adversarial actor and director. The latter provides suggestions for policy perturbations of the protagonist agent to achieve its worst-case performance, while the former generates an adversarial perturbation of observation.

In MARL, cooperation among agents means that an adversary for one protagonist agent must consider its impact on team rewards. The above adversarial training methods to MARL can improve the trained model's robustness (Guo et al., 2024), but there are limitations: 1) **Overfitting to adversarial perturbations**. In MARL, the strongest adversary often causes models to overfit adversarial perturbations of observations, losing performance when handling clean ones due to suboptimal data collection. 2) **Misalignment between actor and director**. When applying the policy adversarial actor and director framework from robust SARL to MARL, there are gaps between the actor and the director. The adversarial perturbations of observations generated by the former do not always align with the latter's intended influence on the protagonist agent's policy, causing instability in training. To address these issues, we propose Adversarial Training with a Stochastic Adversary (ATSA) as a novel framework designed to enhance robustness in MARL with discrete action space and deterministic policies. This study makes the following contributions:

1) It proposes ATSA. Its stochastic adversary consists of two components: a) Stochastic Director (SDor) that suggests policy perturbations by minimizing the expected team reward of the protagonist agent while maximizing entropy to promote exploration and b) SDor-guided generaTor (STor) that generates adversarial perturbations of observations based on SDor's guidance. This framework allows agents to dynamically adjust to both adversarial and clean conditions, thus greatly improving the robustness of MARL models. By leveraging factorized maximum-entropy MARL, we theoretically demonstrate that the soft policy of SDor converges to a global optimum, and when combined with STor, SDor's optimal policy induces the optimal adversary.

2) It designs an SDor-STor loss function, which is new,

to quantify the difference between perturbations suggested by SDor and those implemented by STor. This function is used to refine the coherence between STor and SDor, ensuring that the protagonist agent's policy changes align with the director's intended modifications.

3) To evaluate the robustness of models trained by ATSA, this study conducts extensive experiments on StarCraft II tasks and autonomous driving scenarios. The results show that ATSA is robust to adversarial perturbations of observations generated by four methods while maintaining strong performance in random and perturbation-free environments.

## 2. Preliminaries

In this section, we introduce the definitions of Decentralized Partially Observable Markov Decision Process (Dec-POMDP), Observation-adversarial Dec-POMDP (OD-POMDP), and Policy-adversarial Dec-POMDP (PD-POMDP). Some important concepts, abbreviations, and symbols related to this paper are described in Appendix B.

### 2.1. Dec-POMDP

A Dec-POMDP (Oliehoek & Amato, 2016) is typically described as follows:

$$\mathcal{G} \triangleq \left\langle \mathcal{S}, \{\mathcal{O}^i\}_{i\in\mathcal{N}}, \{\mathcal{A}^i\}_{i\in\mathcal{N}}, \mathcal{N}, r, \{Z^i\}_{i\in\mathcal{N}}, P, \gamma \right\rangle$$

where $\mathcal{S}$ is the state space, $s \in \mathcal{S}$ is a state, $i \in \mathcal{N} \triangleq \{1,\ldots,N\}$ is a protagonist agent, $\mathcal{A}^i$ is the action space for protagonist agent $i$, $a^i \in \mathcal{A}$ is the action taken by protagonist $i$, $\boldsymbol{a} \in \times_{i\in\mathcal{N}}\mathcal{A}^i$ is the joint action of all protagonist agents, $\mathcal{O}^i$ is the observation space of protagonist agent $i$, $o^i \in \mathcal{O}^i$ is the observation of protagonist agent $i$ calculated using the observation function $Z^i(s,i)$ : $\mathcal{S} \times \mathcal{N} \to \mathcal{O}^i$, $r : \mathcal{S} \times_{i\in\mathcal{N}} \mathcal{A}^i \to \mathbb{R}$ is the reward function, $P : \mathcal{S} \times_{i\in\mathcal{N}} \mathcal{A}^i \to \Delta(\mathcal{S})$ is the transition probability function, and $\gamma \in [0,1]$ is the discount factor. Under the Centralized Training with Decentralized Execution (CTDE) paradigm, each protagonist agent learns its policy $\pi^i(a^i|\tau^{i_p}) : \mathcal{T}^{i_p} \times \mathcal{A}^i \to [0,1]$ based on its trajectory $\tau^{i_p} \in \mathcal{T}^{i_p} : \mathcal{O}^i \times \mathcal{A}^i$. From the centralized perspective, the joint policy of all protagonist agents is represented as $\pi^{\mathrm{jt}}(\boldsymbol{a}|\boldsymbol{\tau}^p)$, where $\boldsymbol{\tau}^p \in \times_{i\in\mathcal{N}}\mathcal{T}^{i_p}$ is the joint trajectory of all protagonist agents.

### 2.2. OD-POMDP

An OD-POMDP (Zhou et al., 2024c) extends Dec-POMDP by introducing adversarial perturbations affecting the agents' observations. Its formal definition is as follows:

$$\hat{\mathcal{G}}_{od} \triangleq \left\langle \mathcal{S}, \{\mathcal{O}^i\}_{i\in\mathcal{N}}, \{\mathcal{A}^i\}_{i\in\mathcal{N}}, \{\mathcal{B}^i_\epsilon\}_{i\in\mathcal{M}}, \mathcal{M}, \mathcal{N}, r, \{Z^i\}_{i\in\mathcal{N}}, P, \gamma \right\rangle$$

where $\mathcal{B}_\epsilon^i = \{\hat{o}^i \in \mathcal{O}^i : \|\hat{o}^i - o^i\|_\infty \leq \epsilon\}$ denotes a set of adversarial observation of protagonist agent $i$, an adversarial observation $\hat{o}^i \in \mathcal{B}_\epsilon^i$ is within an $\epsilon$-bounded $\ell_\infty$-distance from the clean observation $o^i$, $\mathcal{M} \subseteq \mathcal{N}$ is the set of observation adversary, and $\hat{\boldsymbol{o}} \triangleq [\hat{o}^i]_{i\in\mathcal{M}}$ is the joint action of observation adversary. The difference between OD-POMDP and Dec-POMDP is the inclusion of observation adversary policy $v^i$ which aims to modify the protagonist's observation. If the observation adversary policy is stochastic, i.e., $v^i(\cdot|\tau^{i_p}) : \mathcal{T}^{i_p} \times \mathcal{B}_\epsilon^i \to [0, 1]$, then $\boldsymbol{v} \triangleq [v^i]_{i\in\mathcal{M}}$ represents the joint policy of the observation adversary, and $\pi_{\boldsymbol{v}}^{\mathrm{jt}}(\cdot|\boldsymbol{\tau}^p) = \pi^{\mathrm{jt}}(\cdot|\tilde{\boldsymbol{\tau}}^p)$, where $\tilde{\boldsymbol{\tau}}^p \triangleq [\tilde{\tau}^i]_{i\in\mathcal{N}}$ is the joint trajectory of protagonist agents, $\tilde{\tau}^{i_p} = \hat{\tau}^{i_p}\mathbb{I}(i \in \mathcal{M}) + \tau^{i_p}\mathbb{I}(i \notin \mathcal{M})$ is the trajectory of the protagonist agent which may be perturbed by adversary, and $\hat{\tau}^{i_p} \in \hat{\mathcal{T}}^{i_p} : \mathcal{B}_\epsilon^i \times \mathcal{A}^i$ is composed of the adversarial observation and the action. $\tilde{\boldsymbol{o}} \sim \boldsymbol{v}(\cdot|\boldsymbol{\tau}^p)$ denotes the joint policy of the protagonist agent under this adversary, where $\tilde{\boldsymbol{o}} \triangleq [\tilde{o}^i]_{i\in\mathcal{N}}$, and $\tilde{o}^i = \hat{o}^i\mathbb{I}(i \in \mathcal{M}) + o^i\mathbb{I}(i \notin \mathcal{M})$.

## 2.3. PD-POMDP

Given a Dec-POMDP $\mathcal{G}$, a fixed joint stochastic policy of protagonist agents $\pi^{\mathrm{jt}}$, and a perturbation budget $\epsilon \geq 0$, a PD-POMDP (Sun et al., 2022b; Zhou & Liu, 2023) is defined as

$$\hat{\mathcal{G}}_{pa} \triangleq \left\langle \mathcal{S}, \{\mathcal{O}^i\}_{i\in\mathcal{M}}, \{\hat{\mathcal{A}}^i\}_{i\in\mathcal{M}}, \mathcal{N}, \mathcal{M}, \hat{r}, \{Z^i\}_{i\in\mathcal{M}}, \hat{P}, \gamma \right\rangle$$

where $\hat{\mathcal{A}}^i : \{d \in [-1, 1]^{|\mathcal{A}^i|}, \sum_{j=1}^{|\mathcal{A}^i|} d_j = 0\}$ is the action space and represents the policy perturbations of the protagonist agent. The relationship between $\hat{P}$ and $P$ (resp. $\hat{r}$ and $r$) in PD-POMDP and Dec-POMDP, respectively, is

$$\hat{P}(s_{t+1}|s_t, \hat{\boldsymbol{a}}_t) = \sum_{\boldsymbol{a}_t \in \mathcal{A}^1 \times \cdots \mathcal{A}^N} \pi^{\mathrm{jt}}(\boldsymbol{a}_t|\boldsymbol{g}(\hat{\boldsymbol{a}}_t, \boldsymbol{o}_t), \tilde{\boldsymbol{\tau}}_{t-1}^p) P(s_{t+1}|s_t, \boldsymbol{a}_t)$$

$$\hat{r}(s_t, \hat{\boldsymbol{a}}_t) = -\sum_{\boldsymbol{a}_t \in \mathcal{A}^1 \times \cdots \mathcal{A}^N} \pi^{\mathrm{jt}}(\boldsymbol{a}_t|\boldsymbol{g}(\hat{\boldsymbol{a}}_t, \boldsymbol{o}_t), \tilde{\boldsymbol{\tau}}_{t-1}^p) r(s_t, \boldsymbol{a}_t).$$

The function $g^i$ generates the corresponding adversarial observations based on the action of the policy adversary $\hat{a}^i$, with $\boldsymbol{g} \triangleq [g^i]_{i\in\mathcal{N}}$. If $i \notin \mathcal{M}$, then $g^i = o_t^i$, and otherwise,

$$g^i(\hat{a}_t^i, o_t^i) = \mathrm{argmax}_{\hat{o}_t^i \in \mathcal{B}_\epsilon^i} \left\| \pi^i\left(\cdot|\hat{o}_t^i, \hat{\tau}_{t-1}^{i_p}\right) - \pi^i\left(\cdot|o_t^i, \hat{\tau}_{t-1}^{i_p}\right) \right\|$$

$$s.t. \left(\pi^i\left(\cdot|\hat{o}_t^i, \hat{\tau}_{t-1}^{i_p}\right) - \pi^i\left(\cdot|o_t^i, \hat{\tau}_{t-1}^{i_p}\right)\right)^T \hat{a}_t^i$$

$$= \left\| \pi^i\left(\cdot|\hat{o}_t^i, \hat{\tau}_{t-1}^{i_p}\right) - \pi^i\left(\cdot|o_t^i, \hat{\tau}_{t-1}^{i_p}\right) \right\| \|\hat{a}_t^i\|$$

$$\tag{1}$$

where $\pi^i(\cdot)$ represents the policy function of the protagonist agent $i$. This optimization seeks the adversarial observation $\hat{o}_t^i$ within the allowable set $\mathcal{B}_\epsilon^i$ that maximally deviates the policy's behavior from the clean observation

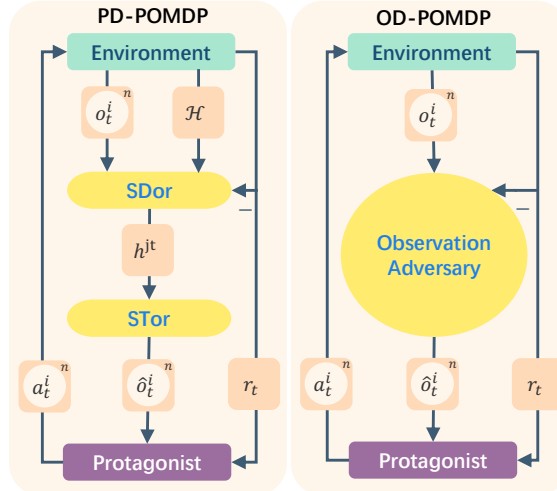

*Figure 1.* The figure illustrates the evolution from OD-POMDP to PD-POMDP. Under $\mathcal{M} = \mathcal{N}$, i.e., all protagonist agents are perturbed by adversaries. In OD-POMDP, the protagonist agent receives observation modified by the observation adversary, makes decisions based on the adversarial observation, and then receives rewards from the environment based on its actions. PD-POMDP generates adversarial observations via SDor and STor. SDor generates policy perturbation suggestions, while STor generates adversarial perturbations based on these suggestions. Unlike OD-POMDP, where the adversary's goal is only to minimize the protagonist agent's reward, PD-POMDP modifies the SDor's objective to include maximizing policy entropy.

$o_t^i$. The constraint ensures that the adversarial observation achieves the maximum possible impact in the direction of $\hat{a}_t^i$. Following Theorem 7 in (Sun et al., 2022b), we know that an optimal policy adversary $h_*^i$ induces an optimal observation adversary against $\pi^i$. In other words, the optimal observation adversary $v_*^i$ in $\hat{\mathcal{G}}_{oa}$ can be constructed by combining $h_*^i$ in PD-POMDP $\hat{\mathcal{G}}_{pa}$ with the function $g^i$. In the CTDE paradigm, the objective of finding an optimal observation adversary $\boldsymbol{v}_*$ in $\hat{\mathcal{G}}_{oa}$ is conducted adversarial training with the protagonist agent, thereby enhancing the robustness of the protagonist agent while this objective can be transformed into the task of finding an optimal joint policy $h^{\mathrm{jt}}$ in $\hat{\mathcal{G}}_{pa}$.

## 3. Adversarial Training with Stochastic Adversary

In this section, we introduce ATSA. As illustrated in Fig. 1, our framework consists of two components: an observation adversary and a protagonist agent.

### 3.1. Objective Function of SDor

In the adversarial training process, using the strongest adversary may lead to instability of training the protagonist

agent or overfitting to the strongest adversarial perturbations, ultimately failing to achieve robustness. To address this problem, we use a stochastic adversary. PD-POMDP $\hat{\mathcal{G}}_{pa}$ is viewed as a probabilistic graphical model. Following (Levine, 2018), we introduce an additional optimality variable $\mathcal{X}_t$ which takes binary values to indicate the optimality of the joint actions taken by all policy adversaries. Specifically, the probability $p(\mathcal{X}_t = 1|\boldsymbol{\tau}_t, \hat{\boldsymbol{a}}_t) \propto \exp(r(s_t, \boldsymbol{a}_t))$ models the likelihood such that a particular joint action is optimal. We define the objective of SDor by using maximum entropy. The related derivation process is given in Appendix C.

**Definition 3.1** (Objective of SDor). The maximum entropy-based objective function of SDor, based on variational inference, is defined as:

$$J\left(h^{\mathrm{jt}}\right) = \mathbb{E}_{s_{1:T} \sim \hat{P}, \hat{\boldsymbol{a}}_{1:T} \sim h^{\mathrm{jt}}}\left[\sum_{t=1}^{T}\left(\hat{r}\left(s_t, \hat{\boldsymbol{a}}_t\right) + \alpha\mathcal{H}\left(h^{\mathrm{jt}}\left(\cdot|\boldsymbol{\tau}_t^a\right)\right)\right)\right] \quad (2)$$

where $\alpha$ is a temperature parameter and $\mathcal{H}(\cdot)$ denotes the entropy of SDor's joint policy.

In Definition 3.1, $\alpha$ controls the trade-off between entropy and reward maximization. As $\alpha \to \infty$, the SDor's policy becomes completely random. By adjusting the value of $\alpha$, the level of randomness in the policy can be effectively controlled. Following the CTDE paradigm, the soft policy of an individual SDor is denoted by $h^i\left(\cdot|\tau^{i_a}\right)$, where $\tau^{i_a} \in \mathcal{T}^{i_a} : \mathcal{O}^i \times \hat{\mathcal{A}}^i$ represents its trajectory. The joint soft policy of all SDors is denoted by $h^{\mathrm{jt}}(\cdot|\boldsymbol{\tau}^a)$, where $\boldsymbol{\tau}^a \in \times_{i \in \mathcal{M}}\mathcal{T}^{i_a}$ represents their joint trajectory. To ensure consistency, we assume that the individual optimal soft policies satisfy the individual-global-optimal condition (Zhang et al., 2021b):

$$h_*^{\mathrm{jt}}\left(\hat{\boldsymbol{a}}|\boldsymbol{\tau}^a\right) = \prod_{i \in \mathcal{M}} h_*^i\left(\hat{a}^i|\tau^{i_a}\right). \quad (3)$$

Then we derive the forms of the optimal joint soft policy and the optimal individual soft policy that maximize this objective. The derivation process is given in Appendix D.

**Proposition 3.2** (Optimal Joint Policy of SDors under Maximum Entropy-based Objective Function). *The optimal joint soft policy of all SDors denoted as $h_*^{jt}$ which maximizes the entropy-regularized objective, is given by:*

$$h_*^{\mathrm{jt}}\left(\hat{\boldsymbol{a}}|\boldsymbol{\tau}^a\right) = \exp\left(\alpha^{-1}\left(Q_*^{\mathrm{jt}}\left(\boldsymbol{\tau}^a, \hat{\boldsymbol{a}}\right) - V_*^{\mathrm{jt}}\left(\boldsymbol{\tau}^a\right)\right)\right), \quad (4)$$

*where $Q_*^{jt}$ is the optimal joint soft Q-function and $V_*^{jt}$ is the optimal joint soft value function.*

**Proposition 3.3** (Optimal Individual Policy of SDor under Maximum Entropy-based Objective Function). *For each SDor $i$, the individual optimal soft policy, which conditions only on its own trajectory $\tau^{i_a}$, is given by:*

$$h_*^i\left(\hat{a}^i|\tau^{i_a}\right) = \exp\left(\alpha_i^{-1}\left(Q_*^i\left(\tau^{i_a}, \hat{a}^i\right) - V_*^i\left(\tau^{i_a}\right)\right)\right), \quad (5)$$

*where $Q_*^i$ and $V_*^i$ are the optimal soft Q-function and value function for the agent $i$, respectively.*

By plugging (4) and (5) into (3), the optimal joint and the individual soft-Q-functions should satisfy

$$Q_*^{\mathrm{jt}}(\boldsymbol{\tau}^a, \hat{\boldsymbol{a}}) = \sum_{i \in \mathcal{M}}\frac{\alpha}{\alpha_i}\left[Q_*^i\left(\tau^{i_a}, \hat{a}^{i_a}\right) - V_*^i\left(\tau^{i_a}\right)\right] + V_*^{\mathrm{jt}}\left(\boldsymbol{\tau}^a\right). \quad (6)$$

## 3.2. SDor and STor

In this section, we provide an explanation of SDor's learning process and STor's computation method. We then theoretically demonstrate that combining SDor and STor results in an optimal stochastic observation adversary.

### 3.2.1. SDOR

We follow the factorized optimal joint policy (Zhang et al., 2021b) to learn the optimal policy of SDor. In the joint soft policy evaluation step, applying a soft Bellman backup operator $\Gamma_{h^{\mathrm{jt}}}$ iteratively to update the joint soft Q-function of SDor $Q_{\mathrm{jt}}$ as $\Gamma_{h^{\mathrm{jt}}}Q^{\mathrm{jt}}\left(\boldsymbol{\tau}_t^a, \hat{\boldsymbol{a}}_t\right) \triangleq \hat{r}_t + \gamma\mathbb{E}_{\boldsymbol{\tau}_{t+1}^a}\left[V^{\mathrm{jt}}\left(\boldsymbol{\tau}_{t+1}^a\right)\right]$ where the soft value function is defined as $V^{\mathrm{jt}}\left(\boldsymbol{\tau}_t^a\right) = \mathbb{E}_{h^{\mathrm{jt}}}\left[Q^{\mathrm{jt}}\left(\boldsymbol{\tau}_t^a, \hat{\boldsymbol{a}}_t\right) - \alpha\log h^{\mathrm{jt}}\left(\hat{\boldsymbol{a}}_t|\boldsymbol{\tau}_t^a\right)\right]$. This formulation provides a foundation for the convergence of the joint soft policy. The individual soft policy is updated according to Proposition 3.3, with more details available in Appendix E. Alternating between joint soft policy evaluation and individual soft policy improvement leads to convergence, as formalized in the following theorem:

**Theorem 3.4.** *(Factorized Soft Policy Iteration of SDor) Consider a joint soft policy of all SDors that can be factorized as $h^{jt} = \prod_{i \in \mathcal{M}} h^i$. By repeatedly applying joint soft policy evaluation and individual soft policy improvement, this process converges to a policy $h_*^{jt}$ such that $Q_{h_*^{jt}}^{jt}(\boldsymbol{\tau}^a, \hat{\boldsymbol{a}}) \geq Q_{h^{jt}}^{jt}(\boldsymbol{\tau}^a, \hat{\boldsymbol{a}})$ for all $[h^i \in \Pi^{h^i}]_{i \in \mathcal{M}}$ and $(\boldsymbol{\tau}^a, \hat{\boldsymbol{a}}) \in \times_{i \in \mathcal{M}}\mathcal{T}^{i_a} \times_{i \in \mathcal{M}} \hat{\mathcal{A}}^i$ with $|\times_{i \in \mathcal{M}} \hat{\mathcal{A}}^i| < \infty$.*

Based on Theorem H.6, we know that the individual soft policies of SDor guarantee convergence to the global optimum. The proof is given in Appendix E. To learn the SDor's soft policies, we employ a deep neural network. During training, the relationship between the joint soft policy and the individual soft policy of SDor is formulated as:

$$Q^{\mathrm{jt}}\left(\boldsymbol{\tau}^a, \hat{\boldsymbol{a}}\right) = \sum_{i \in \mathcal{M}}\lambda_{\phi^a}\left(\boldsymbol{\tau}^a, \hat{\boldsymbol{a}}\right)\left[Q_{\theta^{i_a}}^i\left(\tau^i, \hat{a}^i\right) - V_{\Phi^{i_a}}^i\left(\tau^i\right)\right] + V_{\Phi^a}^{\mathrm{jt}}\left(\boldsymbol{\tau}^a\right) \quad (7)$$

where $\phi^a$, $\Phi^a$, $\Phi^{i_a}$, and $\theta^{i_a}$ represent the parameters of the weight network, the joint soft value network, the individual soft value network, and the individual soft-Q network, respectively. The individual soft-Q network, joint soft value

network, and weight network are trained by minimizing the temporal-difference error:

$$\mathcal{L}\left(\theta^{i_a}, \Phi^a, \phi^a\right) = \mathbb{E}_{(\boldsymbol{\tau}^a, \hat{\boldsymbol{a}}, \hat{r}, \boldsymbol{\tau}^{a'}, \hat{\boldsymbol{a}}') \sim \mathcal{D}^a}\left[\left(Q^{\mathrm{jt}}\left(\boldsymbol{\tau}^a, \hat{\boldsymbol{a}}\right)\right.\right.$$
$$\left.\left. - \left(\hat{r} + \gamma \left(Q_{\mathrm{tar}}^{\mathrm{jt}}\left(\boldsymbol{\tau}^{a'}, \hat{\boldsymbol{a}}' - \alpha \log h^{\mathrm{jt}}\left(\hat{\boldsymbol{a}}'|\boldsymbol{\tau}^{a'}\right)\right)\right)\right)\right)^2\right] \quad (8)$$

where $\mathcal{D}^a$ is the replay buffer of SDor, $Q_{\mathrm{tar}}^{\mathrm{jt}}$ is target network. To train the individual value networks of SDor, the following loss function is minimized:

$$\mathcal{L}\left(\phi^i\right) = \mathbb{E}_{\tau^{ia} \sim \mathcal{D}^a}\left[\left(\mathbb{E}[Q^i(\tau^{ia}, \hat{a}^i) - \alpha_i \log h^i(\hat{a}^i|\tau^i)] - V^i\left(\tau^{ia}\right)\right)^2\right] \quad (9)$$

Finally, the individual soft policies of SDor are optimized using the policy gradient method, where the objective is:

$$J_{h^i}\left(\varphi^i\right) = \mathbb{E}_{\tau^{ia} \sim \mathcal{D}^a, \hat{a}^i \sim h^i}\left[\alpha_i \log h^i\left(\hat{a}^i|\tau^{ia}\right) - Q_{\theta^{ia}}^i\left(\tau^{ia}, \hat{a}^i\right)\right]. \quad (10)$$

### 3.2.2. SТоr

Once the optimal joint policy of SDor is obtained, STor seeks to generate adversarial observations that lead the protagonist agent to take actions aligned with SDor's intentions. The adversarial observation of each protagonist agent can be determined individually based on its clean observations and SDor's actions. The optimization objective for STor $i$ ($i \in \mathcal{M}$) is given in (1). We consider the case where the protagonist agent's policy is deterministic and SDor's policy is stochastic, which modifies the optimization objective function to minimize the following:

$$D_{\mathrm{KL}}\left(\mathbb{E}_{\hat{o}^i \sim h_{\varphi^i}^i}\left[\pi_{\varphi^{i_p}}^i\left(\hat{a}^i|\hat{o}^i, \hat{\tau}^{i_p}\right)\right], h_{\varphi^i}^i\left(\cdot|o^i, \hat{\tau}^{i_p}\right)\right) \quad (11)$$

where $\varphi^{i_p}$ represents the parameters of the policy network of protagonist agent $i$. We then demonstrate that if SDor's joint soft policy is optimal and STor has obtained the optimal solution to (11), this results in the optimal stochastic observation adversary.

**Theorem 3.5** (Optimality of SDor-STor). *For any PD-POMDP $\hat{\mathcal{G}}_{pa}$, any fixed joint deterministic policy of protagonist agents $\pi^{jt}$, and any attack budget $\epsilon \geq 0$, an optimal joint soft policy $h_*^{jt}$ of SDor in $\hat{\mathcal{G}}_{pa}$ induces an optimal an optimal soft policy of the observation adversary against $\pi^{jt}$ in $\hat{\mathcal{G}}_{oa}$.*

Theorem 3.5 demonstrates that if SDor learns the optimal joint soft policy within $\hat{\mathcal{G}}_{pa}$, it can effectively collaborate with STor to produce the optimal stochastic observation adversary. The relevant proof is detailed in Appendix F.

However, (11) is intractable for optimization. In practice, we minimize the expected KL divergence over samples:

$$\mathcal{L}^i\left(\hat{o}^i\right) = \mathbb{E}_{\hat{a}^i}\left[D_{\mathrm{KL}}\left(\pi_{\varphi^{i_p}}^i\left(a^i|\hat{o}^i, \hat{\tau}^{i_p}\right), h_{\varphi^i}^i\left(\hat{a}^i|o^i, \hat{\tau}^{i_p}\right)\right)\right]. \quad (12)$$

We apply the FGSM (Goodfellow et al., 2014) to solve this optimization problem:

$$g^i\left(o^i\right) = \mathrm{clip}\left(o^i - \beta \mathrm{sgn}\left(\nabla_{o^i}\mathcal{L}^i\left(o^i\right)\right)\right) \quad (13)$$

where $\beta$ is the step size, and $\mathrm{clip}(\cdot)$ ensures that the perturbed observation $\hat{o}^i$ remains within a valid range.

### 3.3. SDor-STor Loss Function

From Theorem 3.5, we know that the optimal stochastic observation adversary requires collaboration between SDor and STor. However, previous work (Sun et al., 2022b) does not fully account for this aspect, which may result in decisions by the protagonist agent, based on the adversarial observations generated by STor, that do not align with SDor's intentions. Therefore, we propose an SDor-STor loss function to measure this gap. If the protagonist policy is deterministic and discrete, we use the cross-entropy loss to measure this difference:

$$\mathcal{L}\left(\varphi^i\right) = \mathbb{E}_{\tilde{\tau}^{i_p} \sim \mathcal{D}^p, \tau^{ia} \sim \mathcal{D}^a}\left[\pi^i\left(a^i|\tilde{\tau}^{i_p}\right)\log h^i\left(\hat{a}^i|\tau^{ia}\right)\right] \quad (14)$$

In conjunction with the individual soft policy objective function of SDor (10), we optimize the individual soft policy network by updating it with the gradient of

$$\mathcal{L}'\left(\varphi^i\right) = J\left(\varphi^i\right) + \kappa \mathcal{L}\left(\varphi^i\right) \quad (15)$$

where $\kappa$ is a hyperparameter that regulates the impact of our loss function on SDor's individual policy. For a comprehensive explanation of the training process and complexity analysis, see Appendix G.

## 4. Experiments

### 4.1. Experimental Settings

#### 4.1.1. ENVIRONMENT SETTINGS

We evaluate our adversarial training framework on two challenging benchmarks: the StarCraft Multi-Agent Challenge (SMAC) (Samvelyan et al., 2019) and a Connected and Autonomous Vehicles (CAV) environment (Chen et al., 2023). We conduct experiments on three SMAC maps containing 3 Marines (3m), 3 Stalkers_vs_3 Zealots (3s_3z), and 8 Marines (8m) as well as one CAV scenario with three autonomous vehicles and 1–4 human-driven ones. More details about the SMAC and CAV environments are included in Appendix H.1.

#### 4.1.2. BENCHMARK METHODS

All benchmarks are implemented based on the classical MARL methods: Value Decomposition Network (VDN) (Sunehag et al., 2018) and Q-MIXing network (QMIX) (Rashid et al., 2020). We use the following as benchmarks: No Adversary (NoAdv), Random Noise (RN), adversarial training methods including FGSM (Goodfellow

*Table 1.* Performance comparison in SMAC. The columns represent various adversary algorithms, and the rows indicate the protagonist agent's training methods. AVG represents the average performance of the protagonist agent under six adversaries. The values in the table represent the win rates over 500 episodes (higher is better). The upper bound of the perturbation size is set to 0.25 for 3m and 3s_3z, and 0.1 for 8m. Bold numbers mean the best results.

| Envs | Protagonist | VDN | | | | | | | QMIX | | | | | | |
|---|---|---|---|---|---|---|---|---|---|---|---|---|---|---|---|
| | | NoAdv | RN | FGSM | ATLA | PAAD | ATSA | AVG | NoAdv | RN | FGSM | ATLA | PAAD | ATSA | AVG |
| 3m | NoAdv | 0.99 | 0.23 | 0.00 | 0.02 | 0.00 | 0.00 | 0.21±0.36 | **0.99** | 0.24 | 0.00 | 0.00 | 0.00 | 0.00 | 0.20±0.36* |
| | RAND | **1.00** | **0.99** | 0.01 | 0.94 | 0.01 | 0.01 | 0.49±0.48 | 0.99 | 0.98 | 0.18 | 0.98 | 0.45 | 0.35 | 0.66±0.34* |
| | FGSM | 0.96 | 0.96 | 0.91 | 0.96 | **0.79** | 0.57 | 0.86±0.14 | 0.67 | 0.86 | 0.96 | 0.80 | 0.26 | 0.92 | 0.74±0.24* |
| | ATLA | 0.86 | 0.36 | 0.06 | 0.98 | 0.00 | 0.00 | 0.38±0.40 | **0.99** | 0.68 | 0.00 | 0.98 | 0.00 | 0.00 | 0.44±0.45* |
| | PAAD | 0.93 | 0.94 | 0.87 | 0.96 | 0.81 | 0.89 | 0.90±0.05 | 0.93 | 0.95 | 0.97 | 0.90 | **0.98** | 0.94 | 0.94±0.03 |
| | PR | 0.51 | 0.58 | 0.89 | 0.49 | 0.40 | 0.79 | 0.61±0.17* | 0.80 | 0.84 | 0.91 | 0.75 | 0.73 | 0.62 | 0.78±0.09* |
| | PR-REP | 0.61 | 0.70 | **0.99** | 0.85 | 0.76 | 0.93 | 0.81±0.13 | 0.85 | 0.87 | 0.91 | 0.78 | 0.84 | 0.81 | 0.84±0.04* |
| | ERNIE | 0.83 | 0.87 | 0.95 | 0.77 | 0.73 | 0.37 | 0.75±0.19* | 0.85 | 0.82 | 0.87 | 0.40 | 0.44 | 0.77 | 0.69+-0.19* |
| | RAP | 0.97 | 0.90 | 0.45 | **0.99** | 0.83 | 0.85 | 0.83±0.18 | 0.99 | 0.96 | 0.94 | 0.97 | 0.96 | 0.95 | 0.96±0.02 |
| | ROMANCE-p | 0.86 | 0.31 | 0.00 | 0.09 | 0.00 | 0.00 | 0.21±0.31* | 0.85 | 0.33 | 0.00 | 0.28 | 0.00 | 0.00 | 0.24±0.30* |
| | ROMANCE-s | 0.93 | 0.19 | 0.00 | 0.05 | 0.00 | 0.00 | 0.20±0.34* | 0.98 | 0.42 | 0.00 | 0.01 | 0.00 | 0.00 | 0.23±0.37* |
| | ATSA | 0.97 | 0.95 | 0.96 | 0.84 | 0.77 | **0.98** | **0.91±0.09** | 0.99 | 1.00 | 0.99 | 1.00 | 0.96 | 1.00 | **0.99±0.02** |
| 3s_3z | NoAdv | 0.99 | 0.63 | 0.00 | 0.24 | 0.00 | 0.00 | 0.31±0.38 | **1.00** | 0.76 | 0.00 | 0.49 | 0.00 | 0.00 | 0.38±0.40* |
| | RAND | **1.00** | 0.99 | 0.00 | 0.76 | 0.00 | 0.00 | 0.46±0.46 | **1.00** | **1.00** | 0.00 | 0.95 | 0.00 | 0.00 | 0.49±0.49 |
| | FGSM | 0.92 | 0.93 | 0.68 | 0.84 | 0.36 | 0.38 | 0.69±0.24* | **1.00** | **1.00** | **0.99** | 0.99 | 0.62 | 0.53 | 0.86±0.20 |
| | ATLA | 0.88 | 0.73 | 0.00 | **0.99** | 0.00 | 0.00 | 0.43±0.44 | 1.00 | 0.90 | 0.00 | 1.00 | 0.00 | 0.00 | 0.48±0.48 |
| | PAAD | 0.86 | 0.96 | 0.83 | 0.85 | 0.71 | 0.62 | 0.81±0.11* | 0.97 | 0.98 | 0.33 | 0.97 | 0.42 | 0.45 | 0.69±0.29* |
| | PR | 0.25 | 0.39 | 0.48 | 0.42 | 0.04 | 0.04 | 0.27±0.18* | 0.40 | 0.42 | 0.34 | 0.27 | 0.01 | 0.01 | 0.24±0.17* |
| | PR-REP | 0.96 | 0.98 | 0.91 | 0.84 | 0.63 | 0.52 | 0.81±0.17 | 0.85 | 0.84 | 0.95 | 0.97 | 0.35 | 0.24 | 0.70±0.29* |
| | ERNIE | 0.99 | 0.98 | **0.92** | 0.92 | 0.31 | 0.36 | 0.75±0.29 | **1.00** | **1.00** | 0.98 | **1.00** | 0.83 | 0.84 | 19.11±2.30 |
| | RAP | 0.99 | 0.94 | 0.64 | 0.92 | 0.63 | 0.64 | 0.79±0.16 | 0.98 | 0.98 | 0.77 | 0.95 | 0.78 | 0.70 | 0.86±0.11* |
| | ROMANCE-p | 0.94 | 0.45 | 0.00 | 0.00 | 0.00 | 0.00 | 0.23±0.36* | **1.00** | 0.94 | 0.00 | 0.90 | 0.00 | 0.00 | 0.47±0.47* |
| | ROMANCE-s | 0.97 | 0.41 | 0.00 | 0.05 | 0.00 | 0.00 | 0.24±0.36* | **1.00** | 0.89 | 0.00 | 0.63 | 0.00 | 0.00 | 0.42±0.43* |
| | ATSA | 0.99 | **1.00** | 0.89 | **0.99** | **0.79** | **0.71** | **0.90±0.11** | **1.00** | **1.00** | 0.96 | **1.00** | **0.96** | **0.94** | **0.98±0.02** |
| 8m | NoAdv | 0.96 | 0.86 | 0.00 | 0.34 | 0.00 | 0.00 | 0.36±0.41* | 0.98 | 0.95 | 0.00 | 0.92 | 0.00 | 0.01 | 0.48±0.47* |
| | RAND | 0.98 | 0.98 | 0.02 | 0.91 | 0.01 | 0.05 | 0.49±0.47* | **1.00** | **1.00** | 0.02 | **1.00** | 0.06 | 0.09 | 0.53±0.47 |
| | FGSM | 0.99 | 0.98 | 0.93 | 0.97 | 0.79 | 0.77 | 0.90±0.08* | 0.84 | 0.89 | 0.83 | 0.53 | 0.89 | 0.52 | 0.75±0.16* |
| | ATLA | 0.95 | 0.91 | 0.00 | 0.95 | 0.00 | 0.00 | 0.47±0.47* | 0.95 | 0.94 | 0.02 | 0.95 | 0.00 | 0.00 | 0.48±0.47* |
| | PAAD | **1.00** | **1.00** | 0.94 | 0.99 | **0.92** | 0.92 | 0.96±0.04 | 0.96 | 0.94 | 0.90 | 0.95 | 0.89 | 0.92 | 0.93±0.03* |
| | PR | 0.98 | 0.99 | 0.89 | 0.98 | 0.78 | 0.81 | 0.90±0.08* | 0.96 | 0.95 | 0.93 | 0.97 | 0.47 | 0.09 | 0.73±0.34* |
| | PR-REP | 0.99 | 0.98 | 0.94 | 0.93 | 0.31 | 0.76 | 0.82±0.24* | 0.98 | 0.98 | 0.91 | 0.95 | 0.35 | 0.34 | 0.75±0.29* |
| | ERNIE | 0.99 | 0.98 | **0.95** | 0.95 | 0.90 | 0.84 | 0.94±0.05 | 0.94 | 0.91 | 0.95 | 0.84 | 0.57 | 0.46 | 0.78±0.19* |
| | RAP | 0.97 | 0.98 | 0.93 | 0.96 | 0.79 | **0.94** | 0.93±0.06 | 0.96 | 0.97 | 0.87 | 0.95 | 0.91 | 0.91 | 0.93±0.03* |
| | ROMANCE-p | 0.96 | 0.95 | 0.00 | 0.27 | 0.00 | 0.00 | 0.36±0.43* | 0.98 | 0.99 | 0.08 | 0.99 | 0.03 | 0.02 | 0.52±0.47* |
| | ROMANCE-s | 0.97 | 0.93 | 0.00 | 0.02 | 0.00 | 0.00 | 0.32±0.45* | **1.00** | 0.97 | 0.02 | 0.97 | 0.00 | 0.00 | 0.49±0.49* |
| | ATSA | **1.00** | **1.00** | 0.93 | **1.00** | **0.92** | **0.94** | **0.97±0.04** | **1.00** | **1.00** | **1.00** | **1.00** | **0.97** | **0.95** | **0.99±0.02** |

*\* indicates a statistically significant improvement of ATSA over the corresponding method ($p < 0.05$, Wilcoxon rank-sum test).*

et al., 2014), Alternate Training protagonist agents with Learned Adversaries (ATLA) (Zhang et al., 2021a) and the Policy Adversarial Actor and Director (PAAD) (Guo et al., 2025; Sun et al., 2022b); robust leaning baselines including Policy Regularization (PR) (Guo et al., 2024), Repetitive PR (PR-REP) (Zhou et al., 2024c), and advErsarially Regularized multiageNt reInforcement lEarning (ERNIE) (Bukharin et al., 2023); adversarial training with stochastic adversaries, including Robustness via Adversary Populations (RAP) (Vinitsky et al., 2020), and RObust Multi-AgeNt Coordination via Evolutionary generation (ROMANCE-p/s) (Yuan et al., 2023b). These methods cover a range of adversarial training strategies, from basic noise injection to more sophisticated adversarial training paradigms. Appendix H.2 includes further details on the benchmark methods.

### 4.1.3. PERTURBATION SETTINGS

The perturbation is defined as an $\ell_\infty$ norm with ranges of 0.25 for the 3m and 3s_3z scenarios, 0.1 for the 8m sce-

nario, and 0.05 for the CAV environment. The perturbation range is determined based on the difficulty of the task and the scale of the protagonist agent's observation space. For instance, in the 3m and 3s_3z scenarios, there are fewer agents, resulting in longer distances between agents. Consequently, a larger perturbation range can be used compared to the 8m scenarios. The objective is to select the largest possible perturbation range while maintaining the validity of the task, ensuring the robustness of the model can generalize to a broader range of perturbations.

### 4.1.4. HYPERPARAMETERS

We set the same hyperparameters as those used in the original VDN (Sunehag et al., 2018) and QMIX (Rashid et al., 2020). For SDor, the actor and critic networks consist of two MLP layers with a GRU (hidden size 64) inserted between them. RMSprop (Hinton, 2012; Wen & Zhou, 2024) is used to optimize all parameters, with both actor and critic employing a learning rate of 0.0005. The target networks are updated every 200 episodes. The temperature parame-

*Table 2.* Performance comparison in CAV. The columns represent various adversary algorithms, and the rows indicate the protagonist agent's training methods. AVG denotes the average performance of the protagonist agent across six adversaries. Reward indicates the average cumulative reward over 500 episodes (higher is better), while CR represents the crash rate over the same episodes (lower is better). The upper bound of the perturbation size is 0.05. Bold numbers mean the best results.

| Adversary | | NoAdv | | RN | | FGSM | | ATLA | | PAAD | | ATSA | | AVG | |
|---|---|---|---|---|---|---|---|---|---|---|---|---|---|---|---|
| Protagonist | | Reward | CR | Reward | CR | Reward | CR | Reward | CR | Reward | CR | Reward | CR | Reward | CR |
| VDN | NoAdv | 48.67±29.64 | 0.00 | 51.21±26.23 | 0.00 | 19.51±51.20 | 0.04 | 50.61±28.65 | 0.00 | 22.31±53.24 | 0.07 | 16.91±50.31 | 0.04 | 34.87±15.39* | 0.03±0.03 |
| | RN | 67.31±17.78 | 0.00 | 66.76±19.04 | 0.00 | 39.77±51.37 | 0.05 | 67.24±21.42 | 0.00 | 3.96±69.23 | 0.15 | 17.26±58.26 | 0.08 | 43.72±25.62* | 0.05±0.06 |
| | FGSM | 55.32±35.16 | 0.01 | 58.41±31.77 | 0.01 | 50.53±42.87 | 0.04 | 59.26±29.61 | 0.01 | 33.69±52.99 | 0.05 | 42.96±44.34 | 0.06 | 50.03±9.13* | 0.03±0.02* |
| | ATLA | 57.54±35.72 | 0.02 | 60.23±34.39 | 0.02 | 40.68±58.63 | 0.07 | 61.10±32.54 | 0.01 | -0.67±74.07 | 0.20 | 20.29±64.63 | 0.08 | 39.86±23.11* | 0.07±0.07* |
| | PAAD | 51.18±23.87 | 0.01 | 52.38±22.77 | 0.01 | 46.90±33.86 | 0.01 | 52.97±21.37 | 0.00 | 30.85±48.53 | 0.04 | 30.78±39.58 | 0.03 | 44.18±9.64 | 0.02±0.01 |
| | PR | 62.23±22.41 | 0.00 | 65.24±20.77 | 0.00 | 30.74±62.22 | 0.09 | 66.11±19.07 | 0.01 | -6.95±73.91 | 0.20 | 14.13±67.04 | 0.09 | 38.58±28.17* | 0.07±0.07 |
| | **PR-REP** | 69.78±17.32 | 0.00 | **70.20±16.94** | 0.00 | 61.26±24.82 | **0.00** | 69.74±17.42 | 0.00 | **56.32±35.13** | 0.01 | 52.46±28.74 | 0.00 | **63.29±7.09** | **0.00±0.00** |
| | ERNIE | 67.36±21.72 | 0.00 | 67.73±19.93 | 0.00 | **57.86±29.54** | 0.01 | 67.65±20.72 | 0.00 | 44.49±50.03 | 0.03 | 43.67±34.61 | 0.00 | 58.13±10.51 | 0.01±0.01 |
| | RAP | 66.21±16.93 | 0.00 | 66.26±18.14 | 0.00 | 57.03±32.34 | 0.01 | 66.49±16.95 | 0.00 | 50.08±35.23 | 0.00 | 45.41±31.43 | 0.00 | 58.58±8.44 | 0.00±0.00 |
| | ROMANCE-p | 43.82±35.23 | 0.02 | 43.46±33.36 | 0.00 | 24.17±49.90 | 0.07 | 43.06±34.99 | 0.02 | 7.42±60.47 | 0.14 | -4.21±54.69 | 0.11 | 26.29±19.04* | 0.06±0.05 |
| | ROMANCE-s | **70.49±18.60** | 0.00 | 68.78±24.10 | 0.01 | 24.41±65.56 | 0.11 | **70.38±20.04** | 0.00 | -5.41±73.23 | 0.20 | 1.61±67.03 | 0.11 | 38.38±32.77 | 0.07±0.07 |
| | ATSA | 68.57±22.02 | 0.00 | 68.76±20.54 | 0.00 | 56.05±37.13 | 0.02 | 69.43±19.83 | 0.00 | 40.76±50.01 | 0.05 | **55.32±26.08** | 0.01 | 59.81±10.38 | 0.01±0.02 |
| QMIX | NoAdv | 60.50±30.97 | 0.01 | 61.45±31.71 | 0.02 | 36.48±53.25 | 0.06 | 62.18±29.56 | 0.01 | 16.48±65.48 | 0.12 | 9.83±63.63 | 0.08 | 41.15±21.76* | 0.05±0.04* |
| | RN | **71.49±18.54** | 0.00 | 70.51±22.71 | 0.01 | 23.32±64.68 | 0.07 | 71.32±19.88 | 0.00 | -3.54±68.24 | 0.16 | -2.85±74.44 | 0.13 | 38.38±33.91 | 0.06±0.06 |
| | FGSM | 58.57±27.18 | 0.01 | 59.72±27.12 | 0.01 | 56.72±33.07 | 0.02 | 59.26±28.06 | 0.01 | 46.36±43.27 | 0.03 | 39.30±42.59 | 0.02 | 53.32±7.75* | 0.02±0.01* |
| | ATLA | 49.96±45.70 | 0.02 | 53.21±41.91 | 0.03 | 46.00±50.70 | 0.06 | 49.00±46.74 | 0.04 | 41.14±52.80 | 0.06 | 29.20±57.21 | 0.08 | 44.75±7.89* | 0.05±0.02* |
| | PAAD | 57.43±23.45 | 0.00 | 58.75±25.11 | 0.00 | 47.98±35.80 | 0.02 | 59.36±21.06 | 0.00 | 57.44±23.45 | **0.00** | 35.56±39.43 | 0.01 | 52.75±8.58 | 0.01±0.01 |
| | PR | 59.92±29.28 | 0.01 | 60.88±28.02 | 0.01 | 52.58±36.21 | 0.02 | 60.58±28.08 | 0.01 | 38.92±50.13 | 0.05 | 25.57±57.13 | 0.08 | 49.74±13.26* | 0.03±0.03* |
| | **PR-REP** | 65.52±26.78 | 0.00 | 68.99±19.39 | 0.00 | 61.67±26.62 | 0.01 | 69.79±17.48 | 0.00 | 54.61±26.88 | **0.00** | 51.27±34.29 | **0.00** | 61.98±6.97* | **0.00±0.00** |
| | ERNIE | 59.12±27.78 | 0.00 | 59.72±28.36 | 0.00 | 49.12±36.48 | 0.01 | 59.94±28.08 | 0.00 | 45.98±37.35 | **0.00** | 23.27±46.80 | 0.02 | 49.52±12.95* | 0.01±0.01 |
| | RAP | 69.59± 16.29 | 0.00 | 70.06±15.15 | 0.00 | **63.11±26.19** | **0.00** | **69.85± 16.53** | 0.00 | **59.18±28.33** | 0.01 | 47.04±43.44 | 0.03 | 63.14±8.26 | 0.01±0.01 |
| | ROMANCE-p | 54.93±29.11 | 0.00 | 58.66±26.27 | 0.00 | 32.14±54.22 | 0.08 | 60.00±23.26 | 0.00 | 31.06±53.32 | 0.09 | 12.03±61.42 | 0.13 | 41.47±17.71* | 0.05±0.05 |
| | ROMANCE-s | 63.26±23.82 | 0.00 | 64.69±64.68 | 0.01 | 28.42±58.28 | 0.06 | 65.51±21.92 | 0.00 | 15.66±58.67 | 0.08 | -1.01±61.28 | 0.09 | 39.42±26.48* | 0.04±0.04 |
| | ATSA | 70.17±19.56 | 0.00 | **70.54±18.57** | 0.00 | 62.01±26.66 | **0.01** | 69.80±18.14 | 0.00 | 56.01±31.56 | 0.02 | **53.94±30.66** | 0.01 | **63.74±6.87** | 0.01±0.01 |

*\* indicates a statistically significant improvement of ATSA over the corresponding method (p < 0.05, Wilcoxon rank-sum test).*

ter $\alpha$ and the set $\{\alpha^i\}_{i \in \mathcal{M}}$ follow the same configuration as in (Zhang et al., 2021b). Additionally, our framework introduces an extra hyperparameter, $\kappa$, which regulates the influence of the SDor-STor loss function on the SDor's policy. In our experiments, $\kappa$ for VDN-based protagonist agent is selected from the set $\{0.001, 0.005, 0.01\}$, while for QMIX-based protagonist agent is chosen from the set $\{0.001, 0.005, 0.01, 0.025\}$, depending on the environment to optimize performance.

### 4.2. Results Analysis

The experimental results on SMAC are presented in Table 1, and those on CAV are shown in Table 2. The protagonist model in each cell is trained by using the method specified by the row, while the adversary is trained by using the method specified by the column. Importantly, no retraining is performed for each cell, i.e., the adversary is not retrained specifically to target each protagonist. In SMAC, we use Win Rate (WR) as the metric to evaluate performance. To measure robustness, we calculate the average WR under different perturbations. In CAV, we use reward and Crash Rate (CR) to evaluate model performance, with the average reward and CR under different scenarios also calculated.

For perturbation-free and random noise cases, models trained with FGSM and PAAD show strong performance under adversarial conditions but perform poorly in clean or random noise scenarios. In contrast, our proposed method maintains stable or even improved performance, particularly in CAV, likely due to the soft policy used by adver-

sarial agents, which promotes exploration and reduces local optima risks. Under adversarial observations, NoAdv and RN fail to handle attacks, while RN shows better performance against weaker adversaries like ATLA. PR performs well against FGSM but struggles under stronger attacks like PAAD and ATSA. Among all evaluated frameworks, ATSA achieves the highest average win rate — outperforming 11 baselines across 3 scenarios with 42 statistically significant wins according to the Wilcoxon signed-rank test. Though it doesn't always perform best in every scenario (e.g., 3s_3z with QMIX), it remains generally superior across different adversarial conditions. In CAV, for instance, PR-REP performs well under the VDN framework, but fails under QMIX, indicating difficulty in tuning the regularization weight. Overall, ATSA surpasses most baselines in both cumulative reward and CR. In conclusion: 1) NoAdv and RN fail to enhance robustness. 2) Robust learning methods (PR, PR-REP and ERNIE) face challenges in balancing adversarial and standard losses. 3) Adversarial training methods (FGSM and PAAD) destabilize training under clean conditions; ATLA struggles with large action spaces, reducing its effectiveness. 4) Stochastic adversaries: ROMANCE requires attack budget constraints to stabilize training, which is incompatible with state robustness learning, and RAP avoids constraints via adversary population, but introduces unstable out-of-distribution states and incurs high memory cost. 5) ATSA overcomes overfitting issues by using stochastic adversaries, ensuring strong performance across clean and adversarial conditions. The more detailed experimental results are presented in Appendix H.3. The results and analysis regarding different perturbation ranges are presented in Appendix H.5,

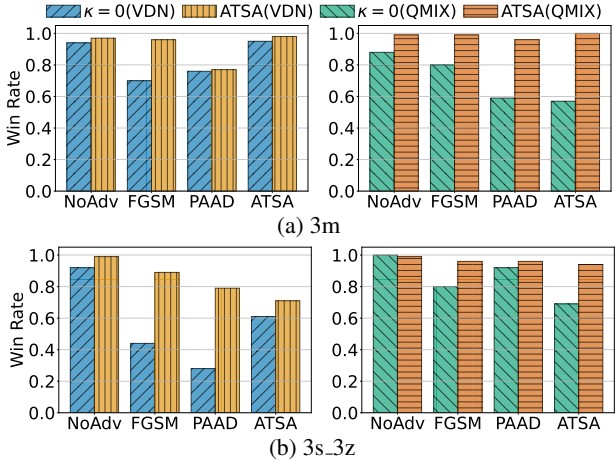

*Figure 2.* Ablation study. The x-axis represents four types of adversaries. The y-axis indicates the win rate of the models under different adversaries. In the legend, $\kappa = 0$ represents the results after removing the SDor-STor loss function.

demonstrating the generalization ability of ATSA from the perspective of perturbation size. Appendix H.6 provides an analysis of the potential for applying ATSA to continuous action spaces.

### 4.3. Ablation Study

Fig. 2 presents the results of the ablation study on the ATSA method, evaluated under two environments: (a) 3m and (b) 3s_3z. $\kappa = 0$ represents the results after removing the SDor-STor loss function. It is evident that the ATSA method demonstrates strong robustness across various adversaries. Particularly in the PAAD and ATSA adversaries, its win rate is significantly higher than that of the $\kappa = 0$ counterparts, highlighting the crucial role in maintaining the consistency between SDor and STor, which ensures more stable training of the adversary. This, in turn, enhances the adversarial robustness of the protagonist agent.

### 4.4. Attacking Performance Analysis

The results in Fig. 3, derived from Table 1, focus on two scenarios: 3m and 3s_3z. Protagonist agents are trained with adversaries like FGSM and PAAD. The comparisons include adversaries trained by ATSA alongside FGSM and PAAD. We selected FGSM and PAAD for comparison because they have shown strong attack performance in prior studies, and models trained using these methods exhibit better robustness against perturbations. Thus, we use ATSA to evaluate its attack capability.

It is important to note that adversaries trained using our method are stochastic, and the optimal adversarial observations they generate adhere to soft constraints. This means the perturbations they create are not always the strongest.

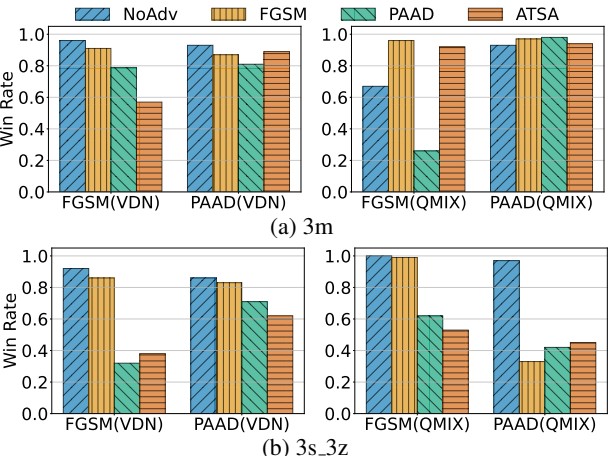

*Figure 3.* Comparison of attack performance across adversaries and scenarios. The x-axis shows models adversarially trained with FGSM and PAAD. NoAdv indicates performance without perturbations. Bars labeled FGSM, PAAD, and ATSA represent win rates under attacks from these methods.

As shown in Fig. 3a, when attacking QMIX-based agents, our method does not always produce the strongest attack results. However, when attacking FGSM(VDN)-based agents, our approach is capable of achieving optimal attack performance. In the 3s_3z scenario, adversaries trained using our framework generally outperform other adversarial attack methods. This is because our method generates more diverse perturbations, thanks to the stochastic nature of the adversaries. For instance, as shown in Fig. 3b, QMIX-based agents trained with FGSM adversarial training show strong defense against FGSM-generated perturbations. However, when attacked by our method, their performance drops significantly. This suggests that models trained with FGSM-based adversarial training tend to overfit to their own perturbation patterns. In summary, our proposed adversarial training framework demonstrates the ability to create adversaries that are effective attackers. These adversaries help expose the weaknesses of existing models, providing a broader perspective for evaluating robustness against adversarial perturbations.

## 5. Conclusion and Future Work

In this paper, we propose an ATSA framework that contains an SDor-STor structure and a novel loss function. ATSA trains the protagonist agent with stochastic adversary, addressing the issues of adversarial observation overfitting and instability in MARL adversarial training. Furthermore, we theoretically prove that in the factorized maximum-entropy framework, the soft policy of SDor converges to a global optimum, and this structure can derive the optimal stochastic observation adversary. To overcome the limitation of prior methods that fail to exploit the policy infor-

mation of the protagonist agents, we introduce the SDor-STor loss function, which leverages the policy information of the protagonist agents when training the adversary policies. Our future work aims to extend the framework to continuous action space algorithms (Lowe et al., 2017; Zhong et al., 2024; Zhou et al., 2024b). Model-based reinforcement learning can also be employed by considering protagonist agents' policies and environments as a unified system. This approach allows the adversary to fully leverage the protagonist agents' information when learning the environment model. ATSA's application to various systems (Hu et al., 2025; Lou et al., 2025) should be pursued.

## Acknowledgements

This work was supported by the National Nature Science Foundation of China under Grant 62172299 and Grant 62032019, and the Fundamental Research Funds for the Central Universities under Grant 2023-4-YB-05.

## Impact Statement

This paper presents ATSA, a method designed to train robust models against diverse observation perturbations. ATSA is applicable to a wide range of real-world scenarios, including autonomous driving and other safety-critical systems.

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

# A. Related Work

## A.1. Multi-Agent Reinforcement Learning

Centralized Training with Decentralized Execution (CTDE) is a popular paradigm to address the challenges of Decentralized Partially Observable Markov Decision Processes (Dec-POMDPs). During the training phase, all agents share global information for centralized training to optimize the overall policy. In the execution phase, each agent independently makes decisions based on its local observations, performing decentralized execution. This paradigm effectively addresses the non-stationarity and partial observability issues in MARL (Foerster et al., 2018; Lowe et al., 2017). Independent deep Q-network (Tampuu et al., 2017) is a simple approach but struggles with complex scenarios due to challenges in reward assignments. Value-based methods like (Rashid et al., 2020; Son et al., 2019; Sunehag et al., 2018) progressively address these limitations by introducing shared rewards, non-linear structures, and broader applicability. More recent methods such as (Wang et al., 2021; 2023; Zhang et al., 2021b) further enhance coordination by leveraging advanced factorization techniques, improving performance across diverse environments and action spaces. However, these classical methods remain sensitive to observation perturbations.

## A.2. Robust Single-Agent Reinforcement Learning

In the field of SARL, improving model robustness relies on two approaches: robust learning and adversarial training. The works in (Liang et al., 2022; Oikarinen et al., 2021; Zhang et al., 2020) are currently the most representative method in robust learning. Zhang et al. (Zhang et al., 2020) develop a policy regularization principle that can be broadly applied to various SARL algorithms. In addition, the State-Adversarial Markov Decision Process (SA-MDP) framework incorporates adversarial perturbations in states, providing a theoretical foundation for robust SARL. In constructing the adversarial regularization term, Liang et al. (Liang et al., 2022) consider not only adversarial perturbations of states but also the significance of the current state within the entire episode. This approach enables the policy network to focus more on the performance of critical states during updates, further enhancing the model's robustness. Unlike the previous two methods, the work in (Oikarinen et al., 2021) employs the concept of interval-bound propagation to construct a regularization term. This approach enhances model robustness by optimizing the boundaries of the decision network. While robust learning can enhance a model's robustness, models trained this way are still sensitive to more potent attacks. Yang et al. (Yang et al., 2024) improves the robustness of reinforcement learning systems by leveraging monitoring mechanisms based on probabilistic automata to guide real-time correction of critical actions . Zhang et al. (Zhang et al., 2021a) use reinforcement learning to Alternate Training protagonist agents with Learned Adversaries (ATLA). This approach performs well in scenarios with small observation spaces. However, as the agent's observation space grows, leading to a larger action space for the adversary, training the adversary model becomes challenging. Sun et al. (Sun et al., 2022b) demonstrate that observation perturbations are equivalent to policy perturbations. Based on this theoretical foundation, the Policy Adversarial Actor and Director (PAAD) is proposed, decomposing the solution of the observation adversary into two steps: the director determines the strongest policy perturbation, and then the actor generates the optimal observation perturbation based on this policy perturbation. This approach effectively addresses the issues present in ATLA. PAAD aims to enhance a model's performance against the strongest adversaries. However, this approach may lead to overfitting to these adversarial perturbations of observations, thereby reducing performance on clean ones.

## A.3. Robust Multi-Agent Reinforcement Learning

In the multi-agent domain, most existing studies primarily focus on action manipulation (Li et al., 2019; 2023; Sun et al., 2022a), policy-level attacks (Guo et al., 2022; Reddi et al., 2024), or adversarial communication (Sun et al., 2023). Other works adopt domain randomization to address the sim-to-real gap (Chen et al., 2024; Shi et al., 2023). In contrast, relatively little attention has been paid to adversarial perturbations on observations. By extending SA-MDP to multi-agent fields, Zhou et al. (Zhou et al., 2024c) propose a state-adversarial stochastic game and discuss its properties and propose a robust training framework based on mean-field actor-critic (Yang et al., 2018), which includes an action loss based on the difference in action distributions between clean and perturbation-free observations, along with the iterative regularization method for action loss. However, this method falls within the scope of robust learning and is also sensitive to stronger adversaries. Guo et al. (Guo et al., 2024; 2025) extend SARL robustness techniques (Goodfellow et al., 2014; Sun et al., 2022b; Zhang et al., 2020; 2021a) to multi-agent scenarios, where multi-agent PAAD achieves notable results. However, it tends to overfit to the perturbation generated by the strongest adversary, compromising performance in other situations. Stochastic adversary has been studied in policy robustness in some papers, such as ROMANCE (Yuan et al., 2023b) and

RAP (Vinitsky et al., 2020). However, there are significant differences between policy robustness and state robustness in terms of both formulation and motivation. Formally, policy robustness assumes that the adversary directly perturbs the agent's actions, modifying the output of the policy. In contrast, state robustness introduces perturbations to the agent's input states. This fundamental difference in formulation directly leads to distinct objectives. In multi-agent settings, policy robustness aims to ensure that the overall system performance remains stable even when some agents' decisions are perturbed. To facilitate stable training under such settings, methods like ROMANCE (Yuan et al., 2023b) introduce a sparse action attack budget to limit the number of adversarial interventions, thereby preserving controllability. In contrast, state robustness focuses on enabling each agent to make correct decisions even when its input states are perturbed. The goal is to make every agent robust to state perturbation. Therefore, the sparse action perturbation constraints used in ROMANCE (Yuan et al., 2023b) are not applicable to state perturbations.

*Table 3.* List of Abbreviations and Their Full Terms

| Abbreviation | Full Term |
|---|---|
| ATLA | Alternate Training Protagonist Agents with Learned Adversaries |
| ATSA | Adversarial Training with Stochastic Adversary |
| CAV | Connected and Autonomous Vehicles |
| CR | Crash Rate |
| CTDE | Centralized Training with Decentralized Execution |
| Dec-POMDP | Decentralized Partially Observable Markov Decision Process |
| FGSM | Fast Gradient Sign Method |
| MARL | Multi-Agent Reinforcement Learning |
| NoAdv | No Adversary |
| OD-POMDP | Observation-adversarial Dec-POMDP |
| PAAD | Policy Adversarial Actor and Director |
| PD-POMDP | Policy-adversarial Dec-POMDP |
| PR | Policy Loss Regularization |
| QMIX | Q-MIXing network |
| RN | Random Noise |
| SARL | Sigle-Agent Reinforcement Learning |
| Sdor | Stochastic Director |
| SMAC | StarCraft Multi-Agent Challenge |
| Stor | SDor-guided generaTor |
| VDN | Value Decomposition Network |
| WR | Win Rate |

# B. Concepts, Abbreviations, and Symbols

## B.1. Concepts

Some important concepts related to this paper are described as follows:

- Protagonist agent: The primary agent whose learning objective is to maximize the cumulative expected reward.

- Clean observations: The accurate observation information directly obtained by an agent from the environment.

- Protagonist policy: The decision-making strategy of the protagonist, designed to guide it in selecting optimal actions based on its observations.

- Observation adversary: An adversary that generates adversarial perturbation to the agent's observations information, aiming to mislead the protagonist agent into making suboptimal decisions.

- Policy adversary: An adversary to modify the protagonist's policy often chooses actions that minimize the protagonist agent's cumulative expected reward.

- Adversarial observations: The observation via adding perturbations generated by the observation adversary to the clean observation.

- Adversarial policy: The decision-making strategy of the policy adversary, specifically designed to disrupt or mislead the protagonist policy.

## B.2. Abbreviations

The abbreviations in the paper are listed in Table 3.

## B.3. Symbols

The symbols in the paper are listed in Table 4.

*Table 4.* Symbols and Explanations

| Category | Symbol | Explanation |
|---|---|---|
| | $\mathcal{S}$ | State space |
| | $s$ | State |
| | $\mathcal{A}^i$ | Action space for protagonist $i$ |
| | $a^i$ | Action taken by protagonist $i$ |
| **Environment** | $\boldsymbol{a}$ | Joint action of all protagonists |
| | $\mathcal{O}^i$ | Observation space of protagonist $i$ |
| | $o^i$ | Observation of protagonist $i$ |
| | $Z^i$ | Observation function |
| | $r$ | Reward function |
| | $P$ | Transition probability function |
| | $\gamma$ | Discount factor |
| | $\mathcal{N}$ | A set of protagonist agent |
| | $\pi^i$ | Policy of protagonist $i$ |
| | $\tau^{i_p}$ | Trajectory of protagonist $i$ |
| | $\pi^{\text{jt}}$ | Joint policy of all protagonists |
| | $\boldsymbol{\tau}^p$ | Joint trajectory of all protagonists |
| | $y^i$ | Action taken by protagonist agent $i$ based on the clean observation |
| **Protagonist Agents** | $\varphi^{i_p}$ | Parameters of the policy network of protagonist agent $i$ |
| | $\pi^{\text{jt}}_{\boldsymbol{v}}$ | Joint policy of protagonist agents under the adversary |
| | $\hat{o}^i$ | Adversarial observation for protagonist $i$ |
| | $\tilde{\boldsymbol{o}}$ | Joint observation of protagonist agents, with some agents perturbed |
| | $\tilde{\boldsymbol{\tau}}^p$ | Joint trajectory of protagonist agents, where some agents are perturbed |
| | $g^i$ | The adversarial observation generated for the agent $i$ based on the action of the policy adversary |
| | $\mathcal{B}^i_\epsilon$ | A set of adversarial observations for protagonist $i$ |
| | $\mathcal{M}$ | Set of observation adversaries |
| | $\hat{\boldsymbol{o}}$ | Joint action of observation adversary |
| **Adversaries** | $v^i$ | Policy of adversary |
| | $\boldsymbol{v}$ | Joint policy of adversary |
| | $\hat{\mathcal{A}}^i$ | Action space for adversary $i$ |
| | $\hat{P}$ | Transition probability function of adversary |
| | $\hat{r}$ | Reward function of adversary |
| | $\mathcal{G}$ | Dec-POMDP |
| **Models and Frameworks** | $\hat{\mathcal{G}}_{oa}$ | OD-POMDP |
| | $\hat{\mathcal{G}}_{pa}$ | PD-POMDP |
| **Learning Parameters** | $\alpha$ | Temperature parameter |
| | $\beta$ | Step size |
| | $\tau^{i_a}$ | SDor's trajectory |
| | $h^{\text{jt}}$ | Joint soft policy of all SDors |
| **SDor's Parameters and Trajectories** | $\boldsymbol{\tau}^a$ | Joint trajectory of all SDors |
| | $Q^{\text{jt}}_*$ | Optimal joint soft Q-function |
| | $V^{\text{jt}}_*$ | Optimal joint soft value function |
| | $\Gamma_{h^{\text{jt}}}$ | Bellman backup operator |
| | $\phi^a$ | Parameters of the weight network |
| **SDor's Network Parameters** | $\Phi^a$ | Parameters of the joint soft value network |
| | $\Phi^{i_a}$ | Parameters of the individual soft value network |
| | $\theta^{i_a}$ | Parameters of the individual soft-Q network |
| **Replay Buffer and Target Networks** | $\mathcal{D}^a$ | Replay buffer of SDor |
| | $Q^{\text{jt}}_{\text{tar}}$ | Target network of joint soft-Q network |

## C. Maximum Entropy-based Objective Function of SDor

The derivation process of Definition 3.1 is as follows.

*Proof:* To derive the maximum entropy objective, we approximate the true distribution of trajectories $\hat{P}(\tau)$, given by

$$\hat{P}(\tau) = \left[ P(s_1) \prod_{t=1}^{T} \hat{P}(s_{t+1}|s_t, \hat{\boldsymbol{a}}_t) \right] \exp\left( \sum_{t=1}^{T} \hat{r}(s_t, \hat{\boldsymbol{a}}_t) \right) \tag{16}$$

with a variational distribution, $\hat{q}(\tau)$, as

$$\hat{q}(\tau) = \hat{q}(s_1) \prod_{t-1}^{T} \hat{q}(s_{t+1}|s_t, \hat{\boldsymbol{a}}_t) \hat{q}(\hat{\boldsymbol{a}}_t|s_t)$$

$$= \hat{q}(s_1) \prod_{t=1}^{T} \hat{q}(s_{t+1}|s_t, \hat{\boldsymbol{a}}_t) \prod_{i=1}^{n} \hat{q}^i(\hat{a}_t^i|s_t) \tag{17}$$

where $\hat{P}(s_{t+1}|s_t, \hat{\boldsymbol{a}}_t)$ and $\hat{q}(s_{t+1}|s_t, \hat{\boldsymbol{a}}_t)$ share the same dynamics, i.e., $P(s_1) = \hat{q}(s_1)$ and $\hat{P}(s_{t+1}|s_t, \hat{\boldsymbol{a}}_t) = q(s_{t+1}|s_t, \hat{a}_t)$, and $\hat{q}(s_{t+1}|s_t, \hat{\boldsymbol{a}}_t) = \prod_{i=1}^{n} \hat{q}^i(\hat{a}_t^i|s_t)$ following the CTDE paradigm.

The variational lower bound on the log-probability of the optimality sequence $\mathcal{X}_{1:T}$ is given by:

$$\log \hat{p}(\mathcal{X}_{1:T}) = \log \int \int \hat{p}(\mathcal{X}_{1:T}, s_{1:T}, \hat{\boldsymbol{a}}_{1:T}) ds_{1:T} d\hat{\boldsymbol{a}}_{1:T}$$

$$= \log \int \int \hat{p}(\mathcal{X}_{1:T}, s_{1:T}, \hat{\boldsymbol{a}}_{1:T}) \frac{\hat{q}(s_{1:T}, \hat{\boldsymbol{a}}_{1:T})}{\hat{q}(s_{1:T}, \hat{\boldsymbol{a}}_{1:T})} ds_{1:T} d\hat{\boldsymbol{a}}_{1:T}$$

$$= \log \mathbb{E}_{(s_{1:T}, \hat{\boldsymbol{a}}_{1:T}) \sim \hat{q}(s_{1:T}, \hat{\boldsymbol{a}}_{1:T})} \left[ \frac{\hat{p}(\mathcal{X}_{1:T}, s_{1:T}, \hat{\boldsymbol{a}}_{1:T})}{\hat{q}(s_{1:T}, \hat{\boldsymbol{a}}_{1:T})} \right]$$

$$\geq \mathbb{E}_{(s_{1:T}, \hat{\boldsymbol{a}}_{1:T}) \sim \hat{q}(s_{1:T}, \hat{\boldsymbol{a}}_{1:T})} \left[ \log p(\mathcal{X}_{1:T}, s_{1:T}, \hat{\boldsymbol{a}}_{1:T}) - \log \hat{q}(s_{1:T}, \hat{\boldsymbol{a}}_{1:T}) \right]. \tag{18}$$

Given that $\hat{q}(s_{t+1}|s_t, \hat{\boldsymbol{a}}) = \hat{p}(s_{t+1}|s_t, \hat{\boldsymbol{a}})$, and substituting (16) and (17) into the bound, we obtain:

$$\log \hat{p}(\mathcal{X}_{1:T}) \geq \mathbb{E}_{(s_{1:T}, \hat{\boldsymbol{a}}_{1:T}) \sim \hat{q}(s_{1:T}, \hat{\boldsymbol{a}}_{1:T})} \left[ \sum_{t=1}^{T} \left( \hat{r}(s_t, \hat{\boldsymbol{a}}_t) - \sum_{i=1}^{n} \log \hat{q}^i(\hat{a}_t^i|s_t) \right) \right]. \tag{19}$$

Optimizing this lower bound with respect to the adversarial policy $\hat{q}(\hat{\boldsymbol{a}}_t|s_t)$ corresponds exactly to the following maximum entropy objective (2). $\blacksquare$

## D. Proof of Propositions 3.2 and 3.3

The derivation process for Propositions 3.2 and 3.3 is the same; here, we only present the derivation for Proposition 3.2.

*Proof:* To derive the optimal joint adversarial policy $h_*^{\text{jt}}$, we start by considering the following constrained policy optimization problem:

$$\max_{h^{\text{jt}}} \mathbb{E}_{\hat{\boldsymbol{a}} \sim h^{\text{jt}}} \left[ Q^{\text{jt}}(\boldsymbol{\tau}^a, \hat{\boldsymbol{a}}) \right] - \alpha \sum_{\hat{\boldsymbol{a}} \in \times_{i \in \mathcal{M}} \hat{\mathcal{A}}^i} h^{\text{jt}}(\hat{\boldsymbol{a}}|\boldsymbol{\tau}^a) \log h^{\text{jt}}(\hat{\boldsymbol{a}}|\boldsymbol{\tau}^a)$$

$$s.t. \sum_{\hat{\boldsymbol{a}} \in \times_{i \in \mathcal{M}} \hat{\mathcal{A}}^i} h^{\text{jt}}(\hat{\boldsymbol{a}}|\boldsymbol{\tau}^a) = 1 \tag{20}$$

To solve this, we construct the Lagrangian:

$$\mathcal{L}(h^{\text{jt}}, \lambda) = \mathbb{E}_{\hat{\boldsymbol{a}} \sim h^{\text{jt}}} \left[ Q^{\text{jt}}(\boldsymbol{\tau}^a, \hat{\boldsymbol{a}}) \right] - \alpha \sum_{\hat{\boldsymbol{a}} \in \times_{i \in \mathcal{M}} \hat{\mathcal{A}}^i} h^{\text{jt}}(\hat{\boldsymbol{a}}|\boldsymbol{\tau}^a) \log h^{\text{jt}}(\hat{\boldsymbol{a}}|\boldsymbol{\tau}^a) - \lambda \left( \sum_{\hat{\boldsymbol{a}} \in \times_{i \in \mathcal{M}} \hat{\mathcal{A}}^i} h^{\text{jt}}(\hat{\boldsymbol{a}}|\boldsymbol{\tau}^a) - 1 \right)$$

$$= \sum_{\hat{\boldsymbol{a}} \in \times_{i \in \mathcal{M}} \hat{\mathcal{A}}^i} h^{\text{jt}}(\hat{\boldsymbol{a}}|\boldsymbol{\tau}^a) Q^{\text{jt}}(\boldsymbol{\tau}^a, \hat{\boldsymbol{a}}) - \alpha \sum_{\hat{\boldsymbol{a}} \in \times_{i \in \mathcal{M}} \hat{\mathcal{A}}^i} h^{\text{jt}}(\hat{\boldsymbol{a}}|\boldsymbol{\tau}^a) \log h^{\text{jt}}(\hat{\boldsymbol{a}}|\boldsymbol{\tau}^a) - \lambda \left( \sum_{\hat{\boldsymbol{a}} \in \times_{i \in \mathcal{M}} \hat{\mathcal{A}}^i} h^{\text{jt}}(\hat{\boldsymbol{a}}|\boldsymbol{\tau}^a) - 1 \right)$$

Taking the derivative of $\mathcal{L}\left(h^{\text{jt}}, \lambda\right)$ with respect to $h^{\text{jt}}\left(\hat{\boldsymbol{a}} | \boldsymbol{\tau}\right)$ and setting it to zero yields:

$$\frac{\partial \mathcal{L}\left(h^{\text{jt}}, \lambda\right)}{\partial h^{\text{jt}}\left(\hat{\boldsymbol{a}} | \boldsymbol{\tau}^a\right)} = Q^{\text{jt}}\left(\boldsymbol{\tau}^a, \hat{\mathbf{a}}\right) - \alpha \log h^{\text{jt}}\left(\hat{\boldsymbol{a}} | \boldsymbol{\tau}^a\right) - \alpha + \lambda = 0 \tag{21}$$

Solving this for the optimal policy $h_*^{\text{jt}}$, we get

$$h^{\text{jt}}\left(\hat{\boldsymbol{a}} | \boldsymbol{\tau}^a\right) = \exp\left(\alpha^{-1}\left[Q^{\text{jt}}\left(\boldsymbol{\tau}^a, \hat{\boldsymbol{a}}\right)\right]\right) \exp\left(\frac{\lambda}{\alpha} - 1\right) \tag{22}$$

Using the normalization condition $\sum_{\hat{\boldsymbol{a}} \in \times_{i \in \mathcal{M}} \hat{\mathcal{A}}^i} h^{\text{jt}}\left(\hat{\boldsymbol{a}} | \boldsymbol{\tau}^a\right) = 1$, we find the optimal $\lambda_*$ as

$$\exp\left(1 - \frac{\lambda_*}{\alpha}\right) = \sum_{\hat{\boldsymbol{a}} \in \times_{i \in \mathcal{M}} \hat{\mathcal{A}}^i} \exp\left(\alpha^{-1} Q^{\text{jt}}\left(\boldsymbol{\tau}^a, \hat{\boldsymbol{a}}\right)\right) \tag{23}$$

Thus,

$$\lambda_* = \left(1 - \log \sum_{\hat{\boldsymbol{a}} \in \times_{i \in \mathcal{M}} \hat{\mathcal{A}}^i} \exp\left(\alpha^{-1} Q^{\text{jt}}\left(\boldsymbol{\tau}, \hat{\boldsymbol{a}}\right)\right)\right) \alpha \tag{24}$$

Substituting $\lambda_*$ back, the optimal policy $h_*^{\text{jt}}$ becomes:

$$h_*^{\text{jt}}\left(\hat{\boldsymbol{a}} | s\right) = \frac{\exp\left(\alpha^{-1} Q_*^{\text{jt}}\left(\boldsymbol{\tau}^a, \hat{\mathbf{a}}\right)\right)}{\sum_{\tilde{\mathbf{a}} \in \times_{i \in \mathcal{M}} \hat{\mathcal{A}}^i} \exp\left(\alpha^{-1} Q_*^{\text{jt}}\left(\boldsymbol{\tau}^a, \tilde{\mathbf{a}}\right)\right)}$$

Since $V_*^{\text{jt}}\left(\boldsymbol{\tau}^a\right) = \alpha \log \sum_{\hat{\boldsymbol{a}} \in \times_{i \in \mathcal{M}} \hat{\mathcal{A}}^i} \exp\left(\alpha^{-1} Q_*^{\text{jt}}\left(\boldsymbol{\tau}^a, \hat{\boldsymbol{a}}\right)\right)$, we conclude that:

$$h_*^{\text{jt}}\left(\hat{\boldsymbol{a}} | \boldsymbol{\tau}^a\right) = \exp\left(\alpha^{-1}\left(Q_*^{\text{jt}}\left(\boldsymbol{\tau}^a, \hat{\boldsymbol{a}}\right) - V_*^{\text{jt}}\left(\boldsymbol{\tau}^a\right)\right)\right) \tag{25}$$

$\blacksquare$

## E. Proof of Theorem 3.4

To prove Theorem H.6, we need the following Lemmas, the proofs of which follow a similar methodology to that presented in (Zhang et al., 2021b).

**Lemma E.1** (Joint Soft Policy Evaluation of SDor). *Consider the soft Bellman backup operator $\Gamma_{h^{\text{jt}}}$ and a mapping $Q_0^{\text{jt}} : \mathcal{S} \times_{i \in \mathcal{M}} \hat{\mathcal{A}}^i \to \mathbb{R}$ with $| \times_{i \in \mathcal{M}} \hat{\mathcal{A}}^i| < \infty$. Define $Q_{k+1}^{\text{jt}} \triangleq \Gamma_{h^{\text{jt}}}$. Then, the sequence $Q_k^{\text{jt}}$ will converge to the joint soft Q-function under the policy $h^{\text{jt}}$ as $k \to \infty$.*

This result guarantees the convergence of the joint soft Q-function for a given joint policy of SDor $h^{\text{jt}}$. Following this, the joint soft policy is updated based on the individual soft policies $h^i{}_{i \in \mathcal{M}}$ under the CTDE paradigm. To ensure consistency and improvement in the individual policies, the following constraint is imposed during the policy update:

$$h_{\text{new}}^i = \arg\min_{h_i'} D_{\text{KL}}\left(h_i'\left(\cdot | \tau^{i_a}\right) \| \exp\left(\alpha_i^{-1}\left(Q_{h_{\text{old}}^i}^i\left(\tau^{i_a}, \cdot\right) - V_{h_{\text{old}}^i}^i\left(\tau^{i_a}\right)\right)\right)\right) \tag{26}$$

This formulation ensures that the new individual policy minimizes the divergence from an exponential transformation of the Q-function. The improvement achieved by this update is formalized in the following lemma:

**Lemma E.2.** *[Individual Soft Policy Improvement of SDor] Let $h_{\text{old}}^i \in \Pi^{h^i}$ and $h_{\text{new}}^i$ be the optimizer of the minimization problem defined in (26). Then, we have $Q_{h_{\text{new}}^{\text{jt}}}^{\text{jt}}\left(\boldsymbol{\tau}_t^a, \hat{\boldsymbol{a}}_t\right) \geq Q_{h_{\text{old}}^{\text{jt}}}^{\text{jt}}\left(\boldsymbol{\tau}_t^a, \hat{\boldsymbol{a}}_t\right)$ for all $\left(\boldsymbol{\tau}_t^a, \hat{\boldsymbol{a}}_t\right) \in \times_{i \in \mathcal{M}} \mathcal{T}^{i_a} \times_{i \in \mathcal{M}} \hat{\mathcal{A}}^i$ with $| \times_{i \in \mathcal{M}} \hat{\mathcal{A}}^i| < \infty$, where $h_{\text{old}}^{\text{jt}} = \prod_{i \in \mathcal{M}} h_{\text{old}}^i$ and $h_{\text{new}}^{\text{jt}} = \prod_{i \in \mathcal{M}} h_{\text{new}}^i$.*

This lemma demonstrates that each iteration of the individual soft policy update improves the value of the joint soft Q-function. By alternating between the joint soft policy evaluation and individual soft policy improvement, the overall factorized soft policy iteration (Zhang et al., 2021b) achieves convergence. This is established in Theorem 3.4. The proof of Theorem H.6 is as follows:

*Proof:* Let $h_k^{\text{jt}}$ represent the soft joint policy at iteration $k$. Based on Lemma E.2, the sequence $Q_{h_k^{\text{jt}}}^{\text{jt}}$ is monotonically increases. Since $Q_{h^{\text{jt}}}^{\text{jt}}$ is bounded above for all $h^{\text{jt}} = \Pi_{i \in \mathcal{M}} h^i$, this sequence converges to some $h_*^{\text{jt}}$.

At convergence, we must have:

$$J_{h_*^{\text{jt}}}\left(h_*^{\text{jt}}\left(\cdot|\boldsymbol{\tau}^a\right)\right) \leq J_{h^{\text{jt}}}\left(h^{\text{jt}}\left(\cdot|\boldsymbol{\tau}^a\right)\right), \forall h^{\text{jt}} \neq h_*^{\text{jt}} \tag{27}$$

Using the same iterative argument as in the proof of Lemma E.2, we conclude that: $Q_{h_*^{\text{jt}}}^{\text{jt}}\left(\boldsymbol{\tau}^a, \hat{\boldsymbol{a}}\right) > Q_{h^{\text{jt}}}^{\text{jt}}\left(\boldsymbol{\tau}^a, \hat{\boldsymbol{a}}\right), \forall\left(\boldsymbol{\tau}^a, \hat{\boldsymbol{a}}\right) \in \times_{i \in \mathcal{M}} \mathcal{T}^{i_a} \times_{i \in \mathcal{M}} \hat{\mathcal{A}}^{i_a}$. Therefore, $h_*^{\text{jt}}$ is optimal in the product space $\Pi_{i \in \mathcal{M}}^i$.

∎

# F. Proof of Theorem 3.5

*Proof:* The logic of the proof is summarized as introducing an intermediate problem $\mathcal{G}_p$ to connect $\hat{\mathcal{G}}_{pa}$ and $\mathcal{G}$, leveraging equivalence and proof by contradiction to ensure the optimality of the policy, thereby validating the theorem's conclusion.

**Definition F.1** (Policy Perturbation Dec-POMDP)**.** For a given Dec-POMDP $\mathcal{G}$, a fixed joint deterministic protagonist policy $\pi^{\text{jt}}$, and an attack $\epsilon$, define a policy perturbation Dec-POMDP as $\mathcal{G}_p \triangleq \langle \mathcal{S}, \{\mathcal{O}^i\}_{i \in \mathcal{M}}, \{\mathcal{A}_p^i\}_{i \in \mathcal{M}}, \mathcal{M}, r_p, \{Z^i\}_{i \in \mathcal{M}}, P_p, \gamma \rangle$, where $\mathcal{A}_p^i = \mathcal{A}^i$, and $\forall s \in \mathcal{S}, a_p^i \in \mathcal{A}_p^i$,

$$r_p(s, \boldsymbol{a}_p) = \begin{cases} -\sum_{\boldsymbol{a}} \boldsymbol{a}_p(\boldsymbol{a}|\boldsymbol{\tau}^p) r(s, \boldsymbol{a}) & \text{if } \boldsymbol{a}_p(\cdot|\boldsymbol{\tau}^p) = \pi^{\text{jt}}(\tilde{\boldsymbol{o}}) \\ -\infty & \text{otherwise} \end{cases} \tag{28}$$

$$P_p(s'|s, \boldsymbol{a}_p) = \sum_{\boldsymbol{a} \in \mathcal{A}} \boldsymbol{a}_p(\boldsymbol{a}|\boldsymbol{\tau}^p) P(s'|s, \boldsymbol{a}) \tag{29}$$

where $\boldsymbol{a}_p$ is the protagonist's policy under policy perturbations and $\mathcal{A} \triangleq \times_{i \in \mathcal{N}} \mathcal{A}^i$ is the joint action space of protagonist. Let $h_p^{\text{jt}}$ denote the optimal joint policy in $\mathcal{G}_p$, We define the entropy-augmented reward as $\tilde{r}_p\left(s_t, \boldsymbol{a}_{p_t}\right) \triangleq r_p\left(s_t, \boldsymbol{a}_{p_t}\right) + \mathbb{E}_{s_{1:t+1} \sim P_p, \boldsymbol{a}_{p_t} \sim h_p^{\text{jt}}}\left[\mathcal{H}\left(h_p^{\text{jt}}\left(\cdot|\boldsymbol{\tau}_t^p\right)\right)\right]$, where $\mathcal{H}_{h_p^{\text{jt}}} \triangleq \mathbb{E}_{s_{1:t+1} \sim P_p, \boldsymbol{a}_{pt} \sim h_p^{\text{jt}}}\left[\mathcal{H}\left(h_p^{\text{jt}}\left(\cdot|\boldsymbol{\tau}_t^p\right)\right)\right]$ then the soft Bellman equation for $h_{p*}^{\text{jt}}$ is given by

$$
\begin{aligned}
V_p^{h_{p*}^{\text{jt}}} &= \max_{h_p^{\text{jt}}} \tilde{r}_p\left(s, h_p^{\text{jt}}(\boldsymbol{\tau}^p)\right) + \gamma \sum_{s' \in \mathcal{S}} P_p\left(s'|s, h_p^{\text{jt}}(\boldsymbol{\tau}^p)\right) V_p^{h_p} \\
&= \max_{h_p^{\text{jt}}} -\sum_{\boldsymbol{a} \in \mathcal{A}} h_p^{\text{jt}}(\boldsymbol{a}|\boldsymbol{\tau}^p) r(s, \boldsymbol{a}) + \gamma \sum_{s' \in \mathcal{S}} \sum_{\boldsymbol{a} \in \mathcal{A}} h_p^{\text{jt}}(\boldsymbol{a}|\boldsymbol{\tau}^p) P(s'|s, \boldsymbol{a}) V_p^{h_p} + \mathcal{H}_{h_p^{\text{jt}}} \\
&= \max_{h_p^{\text{jt}}} \sum_{\boldsymbol{a} \in \mathcal{A}} h_p^{\text{jt}}(\boldsymbol{a}|\boldsymbol{\tau}^p) \left[-r(s, \boldsymbol{a}) + \gamma \sum_{s' \in \mathcal{S}} \sum_{\boldsymbol{a} \in \mathcal{A}} P(s'|s, \boldsymbol{a}) V_p^{h_p}\right] + \mathcal{H}_{h_p^{\text{jt}}}
\end{aligned}
\tag{30}
$$

In the original Dec-POMDP $\mathcal{G}$, an optimal policy adversary $\pi_{\boldsymbol{v}*}^{\text{jt}}$ for $\pi^{\text{jt}}$ is defined as

$$V^{\pi_{\boldsymbol{v}*}^{\text{jt}}} = \min_{\pi_{\boldsymbol{v}}^{\text{jt}}} \sum_{\boldsymbol{a} \in \mathcal{A}} \pi_{\boldsymbol{v}}^{\text{jt}}(\boldsymbol{a}|\boldsymbol{\tau}^p) \left[r(s, \boldsymbol{a}) + \gamma \sum_{s' \in \mathcal{S}} \sum_{\boldsymbol{a} \in \mathcal{A}} P(s'|s, \boldsymbol{a}) V^{\pi_{\boldsymbol{v}}^{\text{jt}}}(s')\right] - \mathcal{H}_{\pi_{\boldsymbol{v}}^{\text{jt}}} \tag{31}$$

By comparing (30) and (31), we have the following lemma,

**Lemma F.2.** *The optimal joint policy in $\mathcal{G}_p$ is an optimal joint policy of observation adversary for $\pi^{jt}$ in $\mathcal{G}$.*

Let $h_*^{\text{jt}}$ denote the optimal joint policy in $\hat{\mathcal{G}}_{pa}$. This policy $h_*^{\text{jt}}$ induces a joint policy $h_p^{\text{jt}_1}$ in $\mathcal{G}_p$. Assume, for contradiction, that $h_p^{\text{jt}_1}$ is not an optimal joint policy in $\mathcal{G}_p$. Then there exists an alternative optimal joint policy, denoted $h_p^{\text{jt}_2}$, in $\mathcal{G}_p$ such that $V_p^{h_p^{\text{jt}_2}}\left(\boldsymbol{\tau}^p\right) > V_p^{h_p^{\text{jt}_1}}\left(\boldsymbol{\tau}^p\right)$ for at least one state $s \in \mathcal{S}$.

---

**Algorithm 1** ATSA

---

Initialize SDor's policy $h^{\text{jt}} \triangleq \{h^i\}_{i \in \mathcal{M}}$ including $\phi^a, \Phi^a, \{\phi^{i_a}, \theta^{i_a}, \varphi^{i_a}\}_{i \in \mathcal{M}}$ and protagonist's policy $\pi^{\text{jt}}$; budget $\epsilon$.

**for** $t = 0, 1, 2, \ldots$ **do**

    For adversary, sample adversarial policy based on SDor's joint soft policy $h^{\text{jt}}$ and generate the adversarial observation based on STor's function (13);

    Protagonist takes action $\boldsymbol{a}_t = \pi^{\text{jt}} \left( \tilde{\boldsymbol{o}}_t^i, \hat{\boldsymbol{\tau}}^p \right)$;

    Put $\langle s_t, \boldsymbol{o}_t, \hat{\boldsymbol{a}}_t, s_{t+1}, \hat{r}_t \rangle$ in the adversary's replay buffer $\mathcal{D}^a$ and $\langle s_t, \tilde{\boldsymbol{o}}_t, \boldsymbol{a}_t, s_{t+1}, r_t \rangle$ in the protagonist's replay buffer $\mathcal{D}^p$;

    **if** Time to update **then**

        Update SDor's network parameters based on (7), (8), (9) and (15);

        Update the protagonist's network parameters based on its algorithm;

    **end if**

**end for**

---

We now construct another joint soft policy $h^{\text{jt}_2}$ in $\hat{\mathcal{G}}_{pa}$ by setting: $h^{\text{jt}_2} \left( \cdot | \boldsymbol{\tau}^a \right) = h_p^{\text{jt}_2} \left( \cdot | \boldsymbol{\tau}^p \right)$. This means $h^{\text{jt}_2}$ follows the same action distribution as $h_p^{\text{jt}_2}$ in $\mathcal{G}_p$. In our framework, the policy $h_p^{\text{jt}_2}$ and $h^{\text{jt}_2}$ are both stochastic, each $\hat{o}_s^i$ satisfies (11), which implies,

$$h^{\text{jt}_2} \left( \cdot | \boldsymbol{\tau}^a \right) = \pi^{\text{jt}} \left( g \left( h^{\text{jt}_2} \left( \cdot | \boldsymbol{\tau}^a \right), \boldsymbol{o}_s \right), \tilde{\boldsymbol{\tau}}^p \right) \tag{32}$$

For the values of the policies, we have:

$$V^{h^{\text{jt}_2}} = \hat{r} + \sum_{s' \in \mathcal{S}} \hat{P} V^{h^{\text{jt}_2}} + \mathcal{H}_{h^{\text{jt}_2}} \tag{33}$$

$$V_p^{h_p^{\text{jt}_2}} = r_p + \sum_{s' \in \mathcal{S}} P_p V_p^{h_p^{\text{jt}_2}} + \mathcal{H}_{h_p^{\text{jt}_2}} \tag{34}$$

Since $h_p^{\text{jt}_2}$ matches $h^{\text{jt}_2}$ in terms of action distributions at each state, we conclude $V_p^{h_p^{\text{jt}_2}} \left( \boldsymbol{\tau}^p \right) = V^{h^{\text{jt}_2}} \left( \boldsymbol{\tau}^a \right)$. Therefore, we have:

$$V^{h^{\text{jt}_2}} \leq V^{h_*^{\text{jt}}} = V_p^{h_p^{\text{jt}_1}} < V_p^{h_p^{\text{jt}_2}} = V^{h^{\text{jt}_2}}$$

This leads to a contradiction, indicating that an optimal soft joint policy in $\hat{\mathcal{G}}_{pa}$ must induce an optimal soft joint policy in $\mathcal{G}_p$. Hence, we conclude the lemma:

**Lemma F.3.** *An optimal soft joint policy in $\hat{\mathcal{G}}_{pa}$ induces an optimal soft joint policy in $\mathcal{G}_p$.*

By combining Lemma F.2 and Lemma F.3, we conclude that an optimal joint policy in $\hat{\mathcal{G}}_{pa}$ induces an optimal stochastic observation adversary for the protagonist $\pi^{\text{jt}}$ in $\mathcal{G}$. Therefore, if SDor learns the optimal joint policy in $\hat{\mathcal{G}}_{pa}$, it can effectively collaborate with STor to generate the optimal stochastic observation adversary.

∎

## G. Complexity Analysis

The training process of ATSA is shown in Algorithm 1. We analyze its overall time complexity.

**Theorem G.1.** *If the time complexity of sampling sample adversarial policy based on SDor's joint soft policy $h^{jt}$ is $D_{p_1}$, computing the adversarial observation based on STor's function (13) is $D_{g_1}$, and taking the joint action of protagonist interacting with the environment is $D_{p_2}$ respectively, the time complexity of sampling process is*

$$D_1 = O \left( T \left( D_{p_1} + D_{p_2} + D_{g_1} \right) \right),$$

*where $T$ is the total training time steps.*

*If the time complexity of computing the gradient of (7), (8), (9) and (15) is $D_{g_2}, D_{g_3}, D_{g_4}$ and $D_{g_5}$ respectively, and the protagonist's network training is $D_{g_6}$, the training time complexity of updating networks is*

$$D_2 = O \left( \frac{T}{T_I} \left( D_{g_2} + D_{g_3} + D_{g_4} + D_{g_5} + D_{g_6} \right) \right),$$

*where $T_I$ is the interval between neural network parameter updates. The time complexity of ATSA is $O\left(D_1 + D_2\right)$.*

## H. Experimental Results

### H.1. Environment settings

We evaluate our adversarial training framework on two challenging benchmarks: the StarCraft Multi-Agent Challenge (SMAC) (Samvelyan et al., 2019) and a Connected and Autonomous Vehicles (CAV) environment (Chen et al., 2023).

SMAC, a distributed real-time strategy game, is a widely adopted benchmark for evaluating MARL performance. It features dynamic, partially observable environments with high-dimensional state and action spaces, requiring agents to coordinate effectively. This makes SMAC an ideal testbed for assessing robustness under adversarial conditions. In contrast, the CAV environment simulates real-world scenarios where connected and autonomous vehicles must collaborate to navigate traffic, avoid collisions, and optimize performance amidst uncertainties. We conduct experiments on three SMAC maps containing 3 Marines (3m), 3 Stalkers_vs_3 Zealots (3s_3z), and 8 Marines (8m) and one scenario on CAV where three autonomous driving vehicles and 1-4 human-driven ones.

In SMAC, we adopt the default episode horizon provided by each environment. The observation space, action space, and reward structure are described as follows.

- Observation space: At each timestep, agents receive localized observations within their field of view. These observations include information about both allies and enemies, such as distance, relative coordinates $(x, y)$, health, shield, and unit type. The environment's global state provides a complete view of all units on the map, including agent positions relative to the map center and other relevant features from their observations. The global state is exclusively used during centralized training. In testing, only the agents' local observations are perturbed in adversarial scenarios.

- Action space: Each agent has four discrete actions available: movement, attacking, stopping, and no-operation. Movement is limited to the four cardinal directions (north, south, east, and west). Agents can attack enemies if they are within range. The stop action leaves the agent idle, while the no-operation (no-op) action is only applicable when the agent is eliminated.

- Reward: The primary objective is to maximize the agents' win rate. SMAC employs a shaped reward system, where agents earn rewards based on the damage they inflict during each timestep. Additional rewards are granted for defeating individual opponents and achieving victory by eliminating all enemy units. All rewards are normalized, ensuring that the maximum cumulative reward for an episode does not exceed 20.

In the CAV environment, we set the episode horizon to 100 timesteps. The corresponding observation space, action space, and reward structure are summarized as follows:

- Observation space: Each agent (vehicle) observes nearby vehicles within a 150-meter range along the longitudinal axis. The observations include features such as whether a vehicle is present, its relative longitudinal and lateral positions $(x, y)$, and its relative longitudinal and lateral speeds $(v_x, v_y)$. Only the nearest neighboring vehicles are considered observable due to local observability constraints.

- Action space: Agents make high-level control decisions, including turning left, turning right, cruising, speeding up, and slowing down. These decisions are translated into low-level steering and throttle commands by the controller. The joint action space of the system is the Cartesian product of all agents' individual action spaces.

- Reward: The reward function incentivizes agents to exhibit desired behaviors, such as safely and efficiently merging into traffic. Rewards are based on several factors: collision avoidance, maintaining stable speeds, minimizing headway time, and successful merging. Each component is weighted to balance the objectives.

### H.2. Benchmark methods

All benchmarks are implemented based on the classical MARL, Value Decomposition Network (VDN) (Sunehag et al., 2018) and Q-MIXing network (QMIX) (Rashid et al., 2020). We use the following methods as the benchmark:

- NoAdv. This baseline uses the standard training approach with VDN and QMIX, without introducing any perturbations during training.

- RN. In this setup, the protagonist is trained with random noise applied to its observations.

- FGSM (Goodfellow et al., 2014). The protagonist is trained with adversarial observation generated using the FGSM. These perturbations are calculated based on the individual protagonist policy, but their impact on the team reward is not considered.

- ATLA (Zhang et al., 2021a). In single-agent scenarios, the observation adversary is trained using Proximal Policy Optimization (PPO) (Schulman et al., 2017), while Multi-Agent PPO (MAPPO) (Yu et al., 2022) is employed as the adversary in multi-agent scenarios. For the MAPPO adversary, clean observations are used as input, adversarial observations are selected as actions, and the reward is defined as the negative of the protagonist's reward. The training alternates between the protagonist and the MAPPO-based observation adversary.

- PAAD (Guo et al., 2025; Sun et al., 2022b). The PAAD adversary has the same network architecture as the protagonist. The adversary operates deterministically on the protagonist's observations and aims to identify its worst-case performance.

- PR (Guo et al., 2024). This robust learning method employs an adversarial policy loss regularization strategy to enhance the robustness of the protagonist models against perturbations. PR formulates the policy loss as $\mathcal{L} = \mathcal{L}_{regular} + \mu \mathcal{L}_{adv}$, where $\mu$ controls the trade-off between standard and adversarial performance.

- PR-REP (Zhou et al., 2024c). $\mu$ is repeatedly increased from 0 to 0.1 three times during training.

- ERNIE (Bukharin et al., 2023). It promotes Lipschitz continuity in policies concerning state observations and actions through adversarial regularization.

- RAP (Vinitsky et al., 2020). It uses a population of adversaries to perturb actions indirectly. To fairly compare with RAP, we have adopted its core idea of promoting adversary diversity by selecting from the population of deterministic adversaries and applying it within the PAAD framework.

- ROMANCE-p/s (Yuan et al., 2023b). This method evolves diverse auxiliary adversarial attackers to introduce policy perturbations during training, thereby enhancing the robustness of multi-agent coordination under uncertainty and potential adversarial threats. We have conducted two sets of experiments: ROMANCE-p, using the original ROMANCE policy adversary, and ROMANCE-s, where our SDor module is replaced with ROMANCE adversaries.

### H.3. Results analysis

For the case of perturbation-free and random noise, the data show that models trained using FGSM and PAAD adversarial frameworks, while performing well under adversarial conditions, tend to exhibit poor performance in clean or random noise scenarios. In contrast, our proposed method does not lead to significant performance degradation under these conditions and even enhances performance in certain clean scenarios, such as in the CAV environment. This improvement may be attributed to the soft policy employed by the adversarial agents in our framework. These stochastic adversaries encourage the protagonist agent to explore a broader range of policies, reducing the likelihood of falling into local optima.

For the case of adversarial observations, the NoAdv and RN baselines do not account for adversarial observations during training, resulting in poor performance under adversarial attacks. However, RN considers random noise during training, allowing it to perform relatively well under weaker adversarial attacks such as those generated by ATLA. PR, a robust regularization-based method, incorporates an adversarial loss term during training. However, its performance under stronger adversarial attacks, such as PAAD and ATSA, remains suboptimal. PR can occasionally achieve the best performance under FGSM attacks, such as in the 8m scenario with QMIX-based methods. The experimental results demonstrate that our proposed ATSA framework achieves the highest average win rate across various scenarios and adversarial observations when compared to FGSM, ATLA, PAAD, and RN methods. While ATSA does not achieve the best performance against all adversaries in every scenario, it demonstrates overall robustness. For example, in the 3s_3z scenario during alternate training with QMIX, ATSA's performance against FGSM as the adversary is not optimal. However, even in this case, the best-performing framework, FGSM, only surpasses ATSA's win rate by 3%, and its performance against other adversaries is not remarkable. A similar pattern is observed in the CAV environment, where ATSA consistently maintains

a robust performance. Moreover, ATSA achieves the best average rewards and collision rates under multiple adversarial conditions, underscoring its superior robustness compared to the other frameworks. From the above analysis, we have the following conclusions:

- NoAdv and RN fail to enhance model robustness under adversarial conditions.

- PR, PR-REP, and ERNIE improve robustness under adversarial conditions by optimizing an adversarial regularization term. However, balancing the adversarial loss with the standard loss proves challenging. This often leads to either a significant performance drop in clean conditions or ineffective adversarial training.

- FGSM and PAAD aim to identify the strongest adversary, optimizing the performance of the protagonist agent under such conditions. While this approach enhances robustness under adversarial observations, it frequently destabilizes training and compromises performance in clean ones. In extreme cases, models trained with these methods may perform equally poorly on both clean and adversarial observations.

- ATLA also seeks to identify the strongest adversary, but the large action space of adversaries makes it difficult to find an optimal solution, reducing its effectiveness in improving robustness.

- ROMANCE as a policy robustness learning method has to constrain adversarial agents, i.e., introducing a sparse action attack budget to limit the number of adversarial interventions, in order to preserve the stability of a training process. These constrains do not work on our state robustness learning method, because if our robustness learning method also takes these constrains, then only a part of agents are possibly robust while others are not.

- RAP as another policy robustness learning method does not use these constraints, and it takes a population of adversaries to deal with overfitting. However, if our state robustness learning method also takes this method to deal with overfitting, lots of out-of-distribution states are generated (so that the training process is not stable) and the memory cost is increased significantly.

- ATSA, our proposed framework, introduces stochastic adversaries to address the overfitting issues inherent in adversarial training. This approach ensures that models achieve strong performance in both clean and adversarial observations by encouraging the protagonist agent to generalize to diverse perturbations.

### H.4. Ablation study

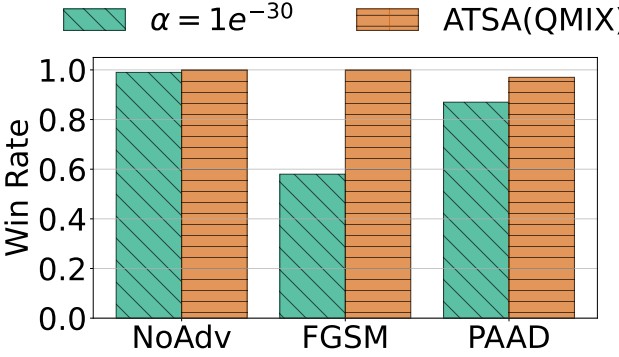

*Figure 4.* Ablation study on $\alpha$ in 8m

The Fig. 4 shows that when $\alpha$ approaches 0, it leads to a decline in robustness. In this case, the policy becomes deterministic, which results in performance similar to that of PAAD—showing good results under PAAD attacks but poor performance under FGSM attacks.

### H.5. Different perturbation range

To test the robustness of the model against perturbations of different ranges, we select the upper bound of perturbation sizes in the range $[0, 0.25]$ with increments of $0.05$. As shown in Fig. 5, we compare the model trained by ATSA with those

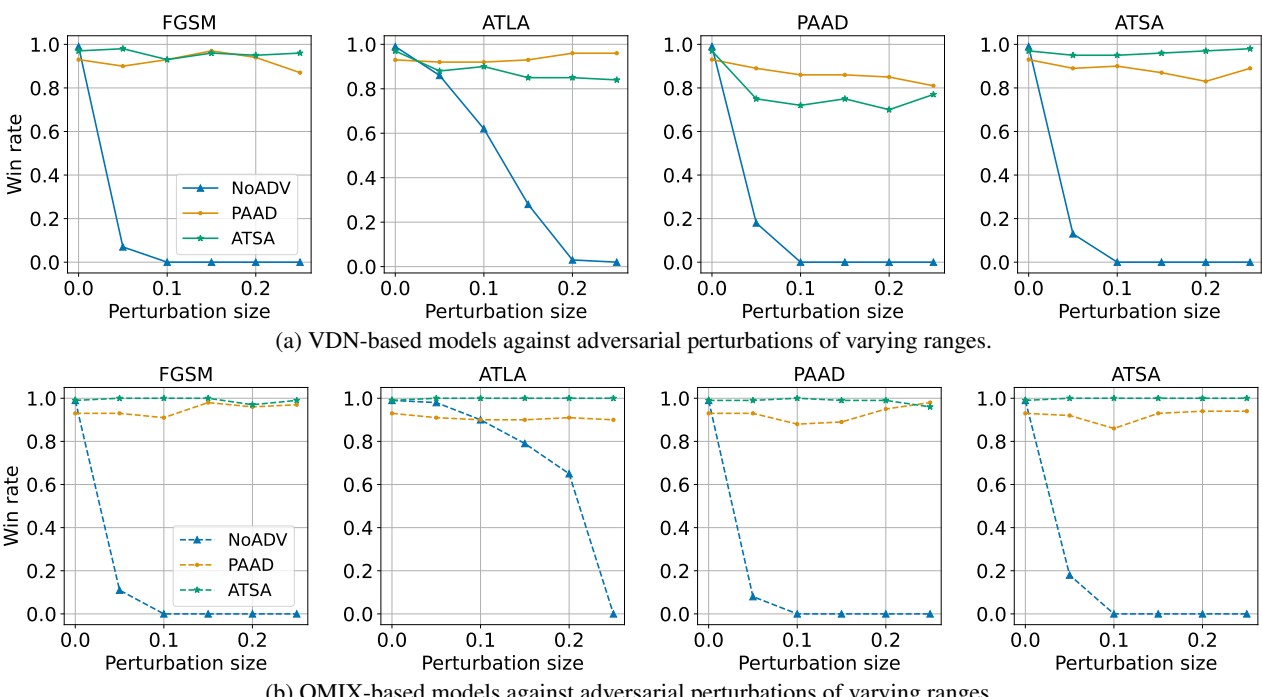

(a) VDN-based models against adversarial perturbations of varying ranges.

(b) QMIX-based models against adversarial perturbations of varying ranges.

*Figure 5.* Robustness evaluation of VDN-based and QMIX-based models against adversarial perturbations of varying ranges. FGSM, ATLA, PAAD, and ATSA are four adversarial attack methods used to evaluate the robustness of models against perturbations. In the legend, NoAdv refers to the baseline model trained without any adversarial training. The models labeled PAAD and ATSA are obtained through adversarial training.

trained by PAAD and NoAdv. We evaluate the model's robustness against perturbations generated using four adversaries focusing on the 3m scenario.

Fig. 5a shows the performance of VDN-based models obtained through training with adversaries. The green solid line represents the model trained using our ATSA method with the VDN framework. For the NoAdv model, its win rate drops to nearly 0 when the perturbation size exceeds 0.05, highlighting its lack of robustness. The only exception is under ATLA attacks, where the performance does not completely degrade due to ATLA's weaker attack capability. Both PAAD and ATSA exhibit stable performance across different perturbation ranges. However, PAAD shows occasional instability, such as at a perturbation size of 0.15, where the performance is significantly better than at 0.05 under the FGSM adversary. This deviation indicates that PAAD-trained models may overfit specific perturbation ranges, leading to unexpected results under varying perturbation sizes.

Fig. 5b depicts the performance of QMIX-based models trained with adversaries. The green dashed line represents the model trained using our ATSA method with the QMIX framework. Under this framework, our method consistently demonstrates superior performance across all attack methods and perturbation ranges. Similar to the observations in Fig. 5a, PAAD-trained models frequently show lower performance under small perturbations than under larger ones. This suggests that PAAD-based training may overfit to specific noise ranges, achieving strong performance within predefined ranges but failing against others. In contrast, our method, which employs stochastic adversaries during adversarial training, successfully counters a diverse set of perturbation-generated methods and ranges. This diversity effectively mitigates the overfitting problem observed in PAAD-trained models.

In summary, our ATSA method, by leveraging stochastic adversarial training, equips the model with robustness against a broader range of perturbation sizes. Compared to PAAD, our approach resolves the overfitting issue, ensuring stable and strong performance across varying perturbation ranges.

## H.6. Continuous action space

*Table 5.* Performance of ATSA in continuous control tasks under the MPE environment

| MPE | NoAdv | RN | FGSM | ATLA | PAAD | ATSA | AVG Reward |
|---|---|---|---|---|---|---|---|
| **MADDPG** | 103.13±68.87 | 110.40±70.05 | 97.88±63.01 | 96.16±66.37 | 107.27±56.33 | 106.67±68.06 | 103.58±5.12* |
| **FGSM** | 121.01±60.14 | 137.47±69.62 | **129.09±63.41** | 115.76±63.58 | 119.60±58.45 | 119.70±62.42 | 123.77±7.32 |
| **ATLA** | 100.81±71.05 | 113.94±75.32 | 98.38±69.64 | 113.13±71.87 | 116.97±71.37 | 115.15±73.21 | 109.73±7.30* |
| **PAAD** | **130.00±68.30** | 118.89±68.64 | 127.88±66.67 | 128.78±71.11 | **121.62±67.54** | 118.38±63.96 | 124.26±4.78 |
| **ATSA** | **130.30±72.24** | **136.36±72.98** | 120.40±67.09 | **130.20±67.37** | 121.62±63.50 | **133.64±71.54** | **128.75±5.87** |
| **FACMAC** | 204.74±90.17 | 193.23±86.40 | 179.29±81.17 | 191.01±89.56 | 171.82±82.32 | 169.20±86.00 | 184.88±12.58 |
| **FGSM** | 138.79±85.03 | 159.90±88.11 | 153.43±70.43 | 151.01±72.73 | 158.98±81.75 | 160.81±78.22 | 153.82±7.60 |
| **ATLA** | 164.34±82.48 | 170.10±76.52 | 160.71±80.41 | 183.54±90.47 | 175.35±93.00 | 158.99±77.58 | 168.84±8.60 |
| **PAAD** | 152.63±76.65 | 150.81±87.62 | 167.27±80.07 | 153.94±75.38 | 155.76±79.81 | 155.86±71.49 | 156.05±5.32 |
| **ATSA** | 166.66±76.19 | 158.99±71.77 | 164.14±84.18 | 159.80±79.42 | 166.46±85.66 | 153.23±69.98 | 161.55±4.76 |

*\* indicates a statistically significant improvement of ATSA over the corresponding method ($p < 0.05$, Wilcoxon rank-sum test).*

To demonstrate the potential applicability of ATSA to continuous control tasks, we conduct experiments based on MAD-DPG (Lowe et al., 2017) and FACMAC (Peng et al., 2021) in predator-prey scenarios where there are 3 agents and 1 prey based on Multiagent Particle Environments (MPE) (Lowe et al., 2017). As shown in Table 5, our method under the MAD-DPG framework shows some improvement compared to existing baselines such as FGSM, PAAD, and ATLA. However, the performance under the FACMAC framework is not particularly strong under both clean and adversarial conditions. This indicates that our method has limitations in continuous action spaces and needs further improvement.

We suspect that this is due to two main reasons: 1) In multi-agent environments, inducing coordinated worst-case behaviors via gradient-based attacks is challenging. 2) For step-wise adversaries like PAAD and ATSA, it is difficult to generate compound perturbations that follow the intended direction of policy disruption. In future work, we plan to further extend our method to more challenging continuous action space environments. This includes improving adversarial optimization techniques for continuous domains and enhancing policy robustness against high-dimensional perturbations.

