# OpenReview forum: "Robust Multi-Agent Reinforcement Learning with Stochastic Adversary"
_ICML.cc/2025/Conference — ICML 2025 poster_

### Official Review · Reviewer_UH2H · 2025-03-12

**Overall Recommendation:** 3

**Summary:**

This paper proposes a soft-policy-based MARL observation adversary consisting of a director module and an attack generator. The convergence guarantee of the director module is provided. Experiments on SMAC and CAV benchmarks demonstrate the effectiveness of the proposed method.

## update after rebuttal

My concerns are largely addressed. I raise my score from 2 to 3. Thank you to the authors for their response and the additional experiments. It is good to see the improvements made to the manuscript. Please make sure the improvements are integrated in the next version.

**Claims And Evidence:**

Yes

**Essential References Not Discussed:**

The purterbation of observations in MARL is an open-MARL problem, where the key factors of the learning process change. A survey of this problem [1] should be discussed. Also, the idea of learning an adversary and proving its optimality is similar with [2], which should be discussed and compared as a baseline.

[1] A Survey of Progress on Cooperative Multi-agent Reinforcement Learning in Open Environment. Lei Yuan, et al. 2023.

[2] Robust Multi-agent Coordination via Evolutionary Generation of Auxiliary Adversarial Attackers. Lei Yuan, Ziqian Zhang, et al. 2023.

**Experimental Designs Or Analyses:**

Yes. I checked Section 4. Experiments.

**Methods And Evaluation Criteria:**

Yes

**Other Comments Or Suggestions:**

None.

**Other Strengths And Weaknesses:**

## Strengths
1. **Rigorous Theoretical Foundation:** The paper provides a thorough theoretical analysis of adversary optimality, including formal proofs for convergence on the adversary's optimal policy. This strengthens the methodological grounding of the approach.
2. **Comprehensive Empirical Evaluation:** The experiments span diverse multi-agent benchmarks and demonstrate consistent improvements over strong baselines.

## Weaknesses
1. **Incremental Algorithmic Contribution:** While the integration of stochastic adversaries is technically sound, the core idea of training robust agents via adversaries is well-explored in prior work. The novelty lies primarily in the specific instantiation rather than a conceptual leap.
2. **Underdeveloped Justification for Stochasticity:** The claim that stochastic adversaries mitigate overfitting is not sufficiently substantiated. Population-based methods like RAP [1] and ROMANCE [2] explicitly address this via diverse adversary ensembles, whereas the paper's stochastic policy approach lacks a direct comparison or analysis to validate this advantage.
3. **Overly Complex Formalisms with Limited Clarity:** The OD-POMDP/PD-POMDP formulations introduce heavy notation without commensurate payoff in insight. Figure 1 occupies too much space but is not clearly explained.

[1] Robust Reinforcement Learning Using Adversarial Populations. Eugene Vinitsky, et al. 2020.

[2] Robust Multi-agent Coordination via Evolutionary Generation of Auxiliary Adversarial Attackers. Lei Yuan, Ziqian Zhang, et al. 2023.

**Questions For Authors:**

See Weaknesses.

**Relation To Broader Scientific Literature:**

The key contributions extend and refine ideas from MARL, robustness in machine learning, and open-environment learning. By addressing the challenge of robustness in MARL, this paper advances these fields and provides a framework for developing more robustly collaborative AI systems.

**Theoretical Claims:**

Yes, I checked Theorem 3.4 and 3.5.

---

> ### Author Rebuttal · Authors · 2025-04-01
>
> Thank you for your insightful suggestion. We have added RAP and ROMANCE as benchmark methods in the 8m VDN-based setting, as shown in Table 7 (https://anonymous.4open.science/r/icml25-9974-8E33/9974.pdf).
>
> Q1: Essential References Not Discussed
>
> We plan to include a discussion of the references in the introduction and incorporate ROMANCE as a baseline method in our experiments to enable a more comprehensive comparison.
>
> Q2: Incremental algorithmic contribution
>
> Although the core idea of training robust agents via adversaries has been explored in prior work, there are still some open and challenging problems, such as overfitting, which has been highlighted in recent papers and is a central focus of our study. Certainly, the concept of stochastic adversaries is not new, as it has been studied in the context of policy robustness in several works, including ROMANCE and RAP. However, there are significant differences between policy robustness and state robustness in terms of both formulation and motivation.
>
> Formally, policy robustness involves perturbing agents' actions to evaluate their robustness, whereas state robustness involves perturbing the agent's input states without directly altering its actions.
>
> ROMANCE, as a policy robustness method, requires constraints on adversarial agents, i.e., introducing a sparse action attack budget to limit the number of adversarial interventions, in order to preserve training stability. These constraints are not applicable to our state robustness training method. If our method takes these constraints, only some agents might become robust, while others would remain sensitive to state perturbation. Our experiments also show this phenomenon.
>
> RAP, as another policy robustness method, does not use these constraints, and it takes a population of adversaries to deal with overfitting. However, if our state robustness learning method adopts a similar approach to address overfitting, it generates numerous out-of-distribution states (resulting in destabilization of the training process) and significantly increases memory costs.
>
> Therefore, although stochastic adversaries have been used for policy robustness, they do not translate well to the setting of state robustness. In contrast, our paper provides a feasible way. We’ll add this discussion in the future version.
>
> Q3: Underdeveloped justification for stochastic adversaries.
>
> We have added both RAP and ROMANCE as baselines in our experiments for comparison.
>
> To fairly compare with RAP, we have adopted its core idea of promoting adversary diversity by selecting from the population of deterministic adversaries and applied it within the PAAD framework. Experimental results show that our method achieves better performance across diverse adversarial scenarios, while being more memory-efficient by noting that RAP requires training and maintaining multiple adversaries in parallel.
>
> For ROMANCE, we have conducted two sets of experiments: ROMANCE-p, using the original ROMANCE policy adversary, and ROMANCE-s, where our SDor module is replaced with ROMANCE adversaries. Results indicate that ROMANCE does not effectively help the protagonist learn state-robust policies.
>
> Q4: Overly complex formalisms with limited clarity
>
> In the definitions, we have used superscripts $p$ and $a$ to distinguish between the states, actions, and policies of the protagonist and the adversary, respectively. We agree that Figure 1 is overly large and under-explained, and we plan to replace it with the linked Figure 2 in the final version.

---

### Official Review · Reviewer_5nJY · 2025-03-16

**Overall Recommendation:** 4

**Summary:**

The paper proposes replacing workst-case adversary to stochastic adversary to improve robustness in multi-agent reinforcement learning (MARL). The proposed adversary model consists of a director and an actor where the former predicts a direction of manipulation and the latter translates this action into a manipulated observation. The paper proves the convergence and optimality of the training dynamics. It also conducts experiments on StarCraft II and autonomous driving datasets that the proposed model effectively address the issues of adversarial overfitting and director-actor misalignment, which are commonly seen in the adversarial MARL literature.

**Claims And Evidence:**

Most claims are supported by clear and convincing evidence, with one perhaps unclear statement:
- The sentence in line 431 starting with "However..." says the proposed ATSA method achieves the **best** attack performance, but Fig. 3(a) PAAD(VDN) seems to imply the ATSA attacker cannot degrade the agent performance as much as the PAAD attacker, right?

**Essential References Not Discussed:**

No missing essential references recognized.

However, the Related Work section only contains discussions with different robust reinforcement learning algorithms. Perhaps some discussions on adversarial learning would help understand the context more. The idea of using randomization to simultaneously detering attacks and maintaining naive-case performance should exist somewhere in the literature.

**Experimental Designs Or Analyses:**

I checked the experiments on metric comparisons, ablation tests, attacker performance, and the perturbation effect. The experiment design is valid and the results support their claim on the proposed advantages.

- Some details on the training parameters is missing, such as epochs, learning rate, horizon, etc., which may influence the reproducibility of the results.

- Some ablation tests on $\alpha$ could make the experiments more complete.

**Methods And Evaluation Criteria:**

Both StarCraft II and Autonomous Driving problems are valid and common problems in MARL. The evaluation metrics are the cumulative rewards and win rate (resp. crash rate) for StarCraft II (resp. Autonomous Driving). The metrics are reasonable.

Benchmarks are a list of other adversarially trained MARL models, as well as the case where the adversay is absent. These benchmarks are also reasonable.

**Other Comments Or Suggestions:**

The last sentence of section 2.2: The notations are a little confusing. Do the authors mean $\hat{o}\sim v(\cdot|\tau)$ instead of  $\tilde{o}\sim v(\cdot|\tau)$ as $v$ is $\mathcal{M}$-dimensional (line 120) and $\tilde{o}$ is $\mathcal{N}$-dimensional (line 128). Also, is it the "joint policy of the protagonist agent" or the joint policy of the adversary?

**Other Strengths And Weaknesses:**

This paper is strong both in originality and technical contents. It innovatively proposed a training approach with stochastic adversaries. This method is itself novel and original and proved effective in ensuring high RL model performance under different types of adversaries. Both theory and experiments are rich enough to support the claimed superiority of the proposed model.

The weakness is in the discrete and finite environment setting. The applicability of ATSA to continuous control (e.g., MADDPG) remains undiscussed.

**Questions For Authors:**

No.

**Relation To Broader Scientific Literature:**

This work belongs to the multi-agent reinforcement learning and adversarial learning literature. The key finding advances the state-of-the-art performance in the robust MARL area.

**Theoretical Claims:**

I checked all proofs and theoretical claims and not spotted any issues, provided that the cited results are correct.

---

> ### Author Rebuttal · Authors · 2025-04-01
>
> Thank you for your question. Based on your suggestion, we have provided additional experimental results, including ablation studies on $\alpha$ (Fig. 1) and continuous control tasks (Table 1), available at the following link: https://anonymous.4open.science/r/icml25-9974-8E33/9974.pdf
>
> Q1: Some unclear statement
>
> The sentence in line 431: You are correct. We have corrected it.
>
> The sentence on line 1096: We define instability as the variation in performance under different perturbation levels. Although ATSA shows a similar trend in this specific figure, the overall results across all eight evaluation plots indicate that ATSA is more stable than PAAD, with consistently smaller performance fluctuations under varying perturbation levels.
>
> The last sentence of section 2.2: Yes, the correct notation should be $\hat o \sim v(\cdot|\tau)$, where $v$ is the joint policy of the adversary.
>
>
> Q2: Training parameters
>
> We run 4,000,000 environment steps across all environments and methods. For SDor, both actor and critic use a learning rate of 0.0005. The target networks are updated every 200 episodes. In SMAC, we use the default episode horizon provided by each environment. For the CAV environment, the horizon is set to 100 timesteps.
>
> Q3: Some ablation tests on $\alpha$.
>
> We have added ablation experiments on the temperature parameter $\alpha$. The results show that when $\alpha$ approaches 0, it leads to a decline in robustness. In this case, the policy becomes deterministic, which results in performance similar to that of PAAD—showing good results under PAAD attacks but poor performance under FGSM attacks.
>
> Q4: Essential References
>
> We plan to add relevant references on adversarial learning and the use of randomization techniques to balance robustness and nominal performance.
>
> Q5: The applicability of ATSA to continuous control (e.g., MADDPG) remains undiscussed.
>
> We have conducted experiments based on MADDPG and FACMAC in the MPE environment to demonstrate the potential applicability of ATSA to continuous control tasks. Our method under the MADDPG framework shows some improvement compared to existing baselines such as FGSM, PAAD, and ATLA. However, the performance under the FACMAC framework is not particularly strong under both clean and adversarial conditions. This indicates that our method has limitations in continuous action spaces and needs further improvement.
>
> We suspect that this is due to two main reasons: 1) In multi-agent environments, inducing coordinated worst-case behaviors via gradient-based attacks is challenging. 2) For step-wise adversaries like PAAD and ATSA, it is difficult to generate compound perturbations that follow the intended direction of policy disruption. We plan to include this discussion in Section 4 of the future version.

---

### Official Review · Reviewer_9h5o · 2025-03-16

**Overall Recommendation:** 3

**Summary:**

This paper proposes Adversarial Training with Stochastic Adversary (ATSA) to fortify the robustness of models trained by multi-agent reinforcement learning. It addresses the overfitting problem of existing methods by training the proposed adversary online alongside the protagonist agent. ATSA implements an SDor-STor structure that performs policy perturbation and uses a loss function that leverages the protagonist agent's policy information when training the adversary policies.

**Claims And Evidence:**

The 1st claim is the mathematical proof proving the SDor’s soft policy converges to a global optimum according to factorized maximum-entropy MARL and leads to the optimal adversary. It is generally supported by the theoretical proof.

The 2nd claim is that the proposed ATSA demonstrates robustness against diverse perturbations of observation while maintaining outstanding performance in environments without perturbation. This is generally supported by the experiments whereas several questions remain, please see below.

**Essential References Not Discussed:**

This is reference is not discussed in the paper.

**Robust Multi-Agent Reinforcement Learning via Adversarial Regularization: Theoretical Foundation and Stable Algorithms. NeurIPS 2023**

**Experimental Designs Or Analyses:**

The experimental design and soundness will be better if more benchmarks are included such as Multi-Agent Particle Environment (MPE). The MARL methods used can be more powerful such as QTRAN.

**Methods And Evaluation Criteria:**

The proposed method is reasonable considering the adversarial training in MARL.

The evaluation is conducted on 2 common benchmarks for MARL the StarCraft and the connected autonomous vehicles, 2 MARL methods (value decomposition network and Q-MIXing network), and 6 baseline agent training methods.

**Other Comments Or Suggestions:**

There is a typo in Table 1's "NoAdv" column - it should be labeled "WR" instead.

Is it necessary for each agent to have an adversarial policy? For example, in a system with 3 agents, could we use only 2 adversarial agent policies that rotate between different combinations of agents? Would such an approach be effective?

**Other Strengths And Weaknesses:**

One of the weakness is the time complexity and the issue of scalability of the proposed method. The introduction of adversarial agents will lead to higher time complexity and limited scalability.

**Questions For Authors:**

How does the stochastic adversary specifically help prevent destabilization of the protagonist agent's training and overfitting to extreme adversarial perturbations? Are the perturbations from the stochastic adversary less extreme compared to other approaches?

In Table 2, under the VDN row and ATLA column, why is the performance (61.10) significantly lower than RN and PR when using the same agent training method and ATLA adversarial attack? Additionally, what explains the large standard deviations in Table 2?

Line 221 states that the considered case involves a deterministic and discrete protagonist agent policy. Given evaluation tasks like StarCraft, how practical is this setting? What are the implications when the policy is non-deterministic?

What specific deep neural network architectures and parameters are used to learn the soft Q-functions and soft policies for SDor?

What performance trade-offs exist between non-adversarial settings and adversarial training when using ATSA?

**Relation To Broader Scientific Literature:**

This paper related to adversarial robustness and multi-agent reinforcement learning.

**Theoretical Claims:**

Read through, but not carefully verify them

---

> ### Author Rebuttal · Authors · 2025-04-01
>
> We have added additional experiments on the MPE environment (Table 1), the QTRAN (Table 4) and non-deterministic policy (Table 2), ERNIE (Table 5), and results for Q5 (Table 6). Please refer to https://anonymous.4open.science/r/icml25-9974-8E33/9974.pdf
>
> Q1: MPE and QTRAN.
>
> As suggested, we have extended our evaluation to include additional benchmarks. The analysis of the MPE experimental results can be found in our response to Reviewer 5nJY, Q5. QTRAN achieves better performance than other methods in the 8m scenario.
>
> Q2: Essential References Not Discussed.
>
> In the final version, we plan to include a discussion of ERNIE in Section 1 and add it as a benchmark method in Section 4. We have conducted a preliminary experiment with ERNIE under the 3m scenario (VDN-based). As seen, although ERNIE performs well in clean and random noise settings, ATSA outperforms ERNIE in average win rate under different types of perturbations.
>
> Q3: Time complexity and scalability
>
> The additional time complexity introduced by adversarial training is indeed inevitable. In the 8m scenario, training VDN with ATSA for 4M steps takes approximately 14 hrs 9 mins, while training with PAAD takes around 13 hrs and 38 mins. This indicates that ATSA incurs a comparable training cost to other adversarial training methods such as PAAD.
>
> Regarding scalability, it should be noted that our method is designed within the CTDE framework, which has inherent scalability limitations. We suspect that challenges remain in both the theoretical convergence of entropy-based Mean-Field RL to the Nash Q-value and in the practical implementation of such methods, which require further investigation.
>
> Q4: Typo
>
> We will correct it in the final version.
>
> Q5: Is it necessary for every agent to have a dedicated adversarial policy?
>
> Yes, it is necessary for each agent to have its own adversarial policy. To test this, we have trained ATSA-based (QMIX) protagonists in the 3m and 8m environments with only two adversarial policies that rotate among different agent combinations at each timestep. We have compared these with settings where all agents are adversarial (i.e., fully coordinated). The results show that using only two adversarial agents in rotation (i.e., ATSA-2) does not consistently improve the robustness of the protagonist.
>
> Q6: How does a stochastic adversary promote stable and generalizable training, and are its perturbations less extreme than those from other methods?
>
> Our method introduces an entropy term with a temperature coefficient $\alpha$ to encourage diverse adversarial behaviors. Early in training, a higher $\alpha$ promotes broader exploration, preventing overly disruptive perturbations and stabilizing learning. As $\alpha$ decays, the adversary focuses more on reward optimization. This controlled exploration leads to less extreme, more varied perturbations, reducing overfitting and training collapse—making our method more stable than others.
>
> Hence yes, compared to other approaches, the perturbations are indeed less extreme.
>
> Q7: VDN+ATLA performance and large standard deviations.
>
> ATLA struggles to learn effective adversarial policies in high-dimensional state spaces, often behaving like random noise. This results in weak protagonist training and degraded performance, even without attacks. The large standard deviations in Table 2 stem from the CAV reward design, where severe penalties for collisions cause high return variance, especially under unstable or adversarial conditions.
>
> Q8: How practical is assuming a deterministic discrete policy in StarCraft, and what if the policy is stochastic?
>
> This setting is widely adopted in the CTDE framework, where many classic multi-agent methods—such as VDN, QMIX, and QTRAN—are also based on discrete action spaces with deterministic policies. Moreover, these methods are commonly evaluated on benchmarks like SMAC/StarCraft, which naturally align with this assumption.
>
> To further explore the applicability in stochastic policy settings, we have also conducted experiments under the non-deterministic policy (i.e., FOP). The preliminary results indicate that our method is also applicable to non-deterministic policies.
>
> Q9: SDor architectures and parameters.
>
> The actor and critic networks of SDor are composed of two MLP layers with a GRU (hidden size 64) inserted between two layers. RMSprop is used to optimize all parameters.
>
> Q10: What performance trade-offs exist between non-adversarial settings and adversarial training when using ATSA?
>
> In our eight experimental settings, ATSA does not achieve the best performance in three cases under the NoAdv condition. However, it outperforms other methods in most settings and achieves the highest average performance across both clean and adversarial conditions.

---

### Official Review · Reviewer_K6AV · 2025-03-17

**Overall Recommendation:** 4

**Summary:**

This paper proposes Adversarial Training with Stochastic Adversary (ATSA) for training Multi-Agent Reinforcement Learning (MARL) models, where the adversary is trained simultaneously with the protagonist agent. ATSA reduces the models' sensitivity to perturbations in observations, while addressing issues with existing adversarial training methods (such as FGSM and PAAD), which tend to overfit to their own perturbations, resulting in models that perform poorly in a no-adversary (clean) setting. This is achieved via a Stochastic Director
(SDor) that performs policy perturbations, and a SDor-guided generator (STor) that generates observation perturbations given SDor's suggestion. SDor is then trained to both minimize the protagonist's reward and maximize the entropy of its policy, with the latter encouraging exploration. The authors also introduce a SDor-STor loss function to quantify and penalize deviations between SDor-suggested perturbations and those produced by STor, thereby aligning STor's perturbations with SDor's intention. Theoretical proofs show that under factorized maximum-entropy MARL, SDor's soft policy achieves a global optimum, resulting in an optimal observation adversary.

Experimental results using an extensive set of baseline techniques (no adversary, random noise, FGSM, ATLA, PR, and PAAD) show that ATSA is robust in a variety of settings. Overall, ATSA achieves the best average performance among all baseline methods, while existing methods such as FGSM and PAAD perform poorly in clean scenarios or against different adversaries. A number of ablation studies are also performed, demonstrating the contribution of the SDor-STor loss function to ATSA's performance.

**Claims And Evidence:**

The paper's claim that ATSA improves robustness is supported by the experimental results, achieving the highest average performance in all the examined scenarios. The claim that existing methods such as FGSM and PAAD overfit to their own perturbations and therefore perform poorly in clean environment and also against other adversarial methods (such as ATSA itself) is also supported by the experimental results. The authors also theoretically prove the optimality of SDor. Ablation studies further justify the use of the SDor-STor loss function and how it leads to higher performance.

The only claims that are somewhat unclear to me are claims 2 and 4 in Section 4.2, please see the relevant question below.

**Essential References Not Discussed:**

None.

**Experimental Designs Or Analyses:**

The experimental design and setup are appropriate for the problem being studied. I have one clarifying question regarding the configuration of different adversaries and training methods, which I've included below.

**Methods And Evaluation Criteria:**

The authors compare against multiple benchmark RL environments, MARL methods, and adversarial/clean settings. The experiments are therefore sound/comprehensive and support the authors' claims.

**Other Comments Or Suggestions:**

1. I suggest the authors perform a statistical test on results in the average columns of Tables 1 and 2, to determine rows where the lower average performance is statistically significant. This could strengthen the paper's claim that ATSA achieves a better "across-the-board" performance.
2. It would also be helpful to clearly state in the introduction that the approach assumes a discrete policy.

**Other Strengths And Weaknesses:**

This is a great paper with a good mixture of novelty, theoretical proofs, and experimental results. I particularly appreciate the depth/diversity of the experimental results, and the ablation study at the end. The overall motivation behind ATSA is insightful and the authors provide a sufficient amount of evidence that their proposed method outperforms state-of-the-art methods.

One limitation of the proposed method is the assumption that the protagonist's policy is deterministic and discrete. However this still covers a large set of problems, and the continuous setting is left for future work.

A slight weakness of the experimental results is that the performance metrics tends to be noisy in some cases, especially for the Connected and Autonomous Vehicles (CAV) environment. This can undermine the statistical significance of the results. However, the fact that ATSA achieves the best average performance across all settings is a strong justification.

**Questions For Authors:**

1. Could you elaborate on how the experiments were conducted for Tables 1 and 2, particularly when pitching one adversary algorithm against a different training method for the protagonist agent? Are models retrained for each cell in the tables (i.e., the protagonist's policy in each cell is trained using the training method specified by the row, but in an environment where the adversary specified by the column is present)?
2. In Tables 1 and 2, are the confidence intervals standard deviations? How many episodes were they computed over?
3. Could you provide more details/justification regarding the following claims in Section 4.2: "PR faces challenges in balancing adversarial and standard losses" and "ATLA struggles with large action spaces, reducing its effectiveness". For the latter, why is ATLA (and not other methods) specifically called out for struggling with large action spaces?

**Relation To Broader Scientific Literature:**

The paper adds to existing literature by designing a novel method that incorporates information about the policy of protagonist agents, addressing a limitation of prior techniques that do not perform well in scenarios other than their own. This ultimately results in more robust MARL models that perform optimally in a more diverse set of conditions.

**Theoretical Claims:**

I did not check theoretical proofs.

---

> ### Author Rebuttal · Authors · 2025-04-01
>
> Thank you for the suggestion. Additional results—including continuous (Table 1), non-deterministic (Table 2) cases, and further details for Q5 (Tables 1 and 3)—are available at the following link: https://anonymous.4open.science/r/icml25-9974-8E33/9974.pdf
>
> Q1: Deterministic and discrete protagonist policy limits its applicability.
>
> We have now added experiments on continuous action spaces and non-deterministic policies. In continuous action space tasks, ATSA demonstrates a certain level of effectiveness, but there is room for improvement. Please refer to the response to Reviewer 5nJY, Q5 for more details. In the non-deterministic setting (i.e., FOP), our method outperforms baselines. However, our theoretical analysis assumes a deterministic and discrete policy, and extensions to other settings remain future work. We plan to explicitly state this assumption in the introduction and provide further discussion in the future work section.
>
> Q2: Statistical significance tests.
>
> We have conducted Wilcoxon Signed-Rank Tests to assess the statistical significance of ATSA's performance. In Table 1, ATSA shows significant improvement in 24 out of 36 baseline comparisons. In Table 2, ATSA significantly outperforms baselines in 10 out of 12 cases for Reward and 9 out of 12 for CR.
>
> We plan to mark significant results (p < 0.05) with an asterisk (*) in the AVG columns of Tables 1 and 2 in the final version.
>
> Q3: Are the protagonist models retrained for each cell in Tables 1 and 2 based on the specified training method and adversary, or is a shared model used across settings?
>
> The protagonist model in each cell is trained by using the method specified by the row, while the adversary is trained by using the method specified by the column. Importantly, no retraining is performed for each cell, i.e., the adversary is not retrained specifically to target each protagonist. For example, when evaluating PAAD (as the protagonist) against ATSA (as the adversary), the PAAD model is trained independently using its own adversarial training procedure, and the ATSA adversary is also trained independently using its own procedure—not against the PAAD-trained policy specifically. Similarly, the PAAD protagonist is not trained using the ATSA adversary. We will clarify this setup in Section 4.1.2 of the future version.
>
> Q4: Do the confidence intervals in Tables 1 and 2 represent standard deviations, and over how many episodes are they computed?
>
> Yes, the values are standard deviations computed over 500 evaluation episodes.
>
> Q5: More justification for the claims that (1) PR struggles to balance adversarial and standard losses, and that (2) ATLA is less effective in large action spaces.
>
> (1) PR formulates the policy loss as $\mathcal L = \mathcal L_{regular} + \mu \mathcal L_{adv}$, where $\mu$ controls the trade-off between standard and adversarial performance.
>
> Following [1], we set $\mu=0$ for the first 2M steps and $\mu=0.1$ thereafter (denoted PR-0.1). To further investigate this trade-off, we have run two additional experiments: PR-0.05, where $\mu=0.05$ in the final 2M steps and Rep-PR [2], where $\mu$ is repeatedly increased from 0 to 0.1 three times during training. Results on the 3m (QMIX-based) setting are available at the provided link.
>
> These results show that small changes in $\mu$ have notable impacts. While Rep-PR helps on clean inputs, it is less robust under attack. This illustrates the difficulty PR faces in balancing clean and adversarial performance. We plan to include Rep-PR as a benchmark method and report full results in the future version.
>
> (2) ATLA models the adversary as an RL agent that learns to perturb the environment state. This means the adversary's action space is tied to the dimensionality of the state space. As stated in [3]: “If the original state space is high-dimensional, learning a good policy in the adversary’s MDP (i.e., ATLA) may become computationally intractable.”
>
> To illustrate this, we have conducted an experiment in the MPE environment, where the state dimensionality is relatively small (16). Based on FACMAC, the results of ATLA and PAAD-trained protagonists against PAAD are 175.35 and 155.76.
>
> These results show that ATLA performs well in low-dimensional settings. However, in our main experimental environments, the state dimension ranges from 30 to 80, which significantly increases the adversary's action space and training complexity. As a result, ATLA becomes less effective. This supports our claim that ATLA struggles with large state spaces, reducing its effectiveness in high-dimensional environments.
>
> [1] Enhancing the robustness of QMIX against state-adversarial attacks, 2024
>
> [2] A Robust Mean-Field Actor-Critic Reinforcement Learning Against Adversarial Perturbations on Agent States, 2024
>
> [3] Who is the strongest enemy? Towards optimal and efficient evasion attacks in deep RL, 2022

---

### Decision · Program_Chairs · 2025-05-01

**Decision:**

Accept (poster)

**Comment:**

The paper proposes the use of stochastic adversary to improve robustness of multi-agent reinforcement learning (MARL).  Both theoretical and experimental contributions are good.  The reviews are generally positive. The reviewer of the original low score (2) was happy with the rebuttal and expressed willingness to raise their score.